# Nemo-like kinase is a novel regulator of spinal and bulbar muscular atrophy

Tiffany W Todd[1], Hiroshi Kokubu[1], Helen C Miranda[2], Constanza J Cortes[2], Albert R La Spada[2], Janghoo Lim[1]*

[1]Program in Cellular Neuroscience, Neurodegeneration and Repair, Department of Genetics, Yale School of Medicine, New Haven, United States; [2]Departments of Cellular and Molecular Medicine, Neurosciences, and Pediatrics, Division of Biological Sciences, Institute for Genomic Medicine, Sanford Consortium for Regenerative Medicine, University of California, San Diego, La Jolla, United States

**Abstract** Spinal and bulbar muscular atrophy (SBMA) is a progressive neuromuscular disease caused by polyglutamine expansion in the androgen receptor (AR) protein. Despite extensive research, the exact pathogenic mechanisms underlying SBMA remain elusive. In this study, we present evidence that Nemo-like kinase (NLK) promotes disease pathogenesis across multiple SBMA model systems. Most remarkably, loss of one copy of *Nlk* rescues SBMA phenotypes in mice, including extending lifespan. We also investigated the molecular mechanisms by which NLK exerts its effects in SBMA. Specifically, we have found that NLK can phosphorylate the mutant polyglutamine-expanded AR, enhance its aggregation, and promote AR-dependent gene transcription by regulating AR-cofactor interactions. Furthermore, NLK modulates the toxicity of a mutant AR fragment via a mechanism that is independent of AR-mediated gene transcription. Our findings uncover a crucial role for NLK in controlling SBMA toxicity and reveal a novel avenue for therapy development in SBMA.

*For correspondence: janghoo.
lim@yale.edu

**Reviewing editor**: Louis Ptáček, University of California, San Francisco, United States

## Introduction

Spinal and bulbar muscular atrophy (SBMA; MIM #313200) is an X-linked progressive neuromuscular disease (*Kennedy et al., 1968*). Patients present in midlife with weakness of the limb and facial muscles, the latter of which often progress to dysarthria and dysphagia, occasionally leading to fatality. SBMA patients also commonly suffer from mild androgen insensitivity, presenting with gynecomastia, testicular atrophy, and decreased fertility (*Katsuno et al., 2012*). SBMA was originally defined as a neurodegenerative disease affecting the proximal spinal and bulbar motoneurons, and muscle atrophy was considered secondary to motoneuron degeneration. Current opinion in the field of SBMA research, however, now favors a model in which SBMA also directly affects the skeletal muscles (*Yu et al., 2006*; *Jordan and Lieberman, 2008*; *Monks et al., 2008*; *Boyer et al., 2013*; *Malena et al., 2013*; *Oki et al., 2015*), and, in fact, recent studies have shown that removing or decreasing the expression of the mutant protein within skeletal muscle is sufficient to rescue SBMA phenotypes in vivo (*Cortes et al., 2014*; *Lieberman et al., 2014*). This model of disease is supported by the finding that, in conjunction with neuronal loss, patients also show elevated creatine kinase levels and evidence of myopathic changes on biopsy (*Sorarù et al., 2008*; *Chahin and Sorenson, 2009*).

SBMA is caused by the expansion of a polymorphic CAG trinucleotide repeat located in the first exon of the *Androgen Receptor* (*AR*) gene (*La Spada et al., 1991*). In wild-type AR, this repeat region encodes a stretch of 6–36 glutamines (Q). In SBMA patients, in contrast, the region is expanded to 37 to 70Q, resulting in pathogenesis via a gain-of-function and partial loss-of-function mechanism (*Katsuno et al., 2012*). SBMA is therefore one of nine identified polyglutamine (polyQ) repeat

**eLife digest** Spinal and bulbar muscular atrophy (SBMA) is an inherited disease that eventually leads to degeneration in motor neurons and weakness in muscles. It is caused by a specific genetic mutation in the gene that encodes the androgen receptor protein, which leads to the production of a mutant protein that is larger than normal. Similar mutations in other genes can lead to the development of other so-called 'polyglutamine' diseases such as Huntington's disease and spinocerebellar ataxia. However, the precise details of how these mutations lead to disease symptoms are not known, and there are currently no effective ways of treating these conditions.

Previous research has shown that an enzyme called Nemo-like kinase (or NLK for short) regulates the normal androgen receptor in cancer cells. NLK has kinase activity, that is, it adds phosphate molecules to other proteins to regulate their activity. Todd et al. used human cells, fruit flies, and mice as model systems to investigate whether NLK is involved in the development of SBMA.

The experiments show that NLK promotes the development of features associated with SBMA in all three models. The kinase activity of NLK is required for these features to develop. Todd et al. also found that NLK can bind to and add phosphate molecules to the mutant version of the androgen receptor protein. This causes the mutant androgen receptor proteins to accumulate and increases the ability of the mutant proteins to activate particular genes.

Todd et al.'s findings suggest that NLK promotes the development of SBMA by interacting with the mutant androgen receptor. Previous studies have shown that NLK is able to modulate the development of spinocerebellar ataxia type 1, which suggests that NLK may also play an important role in other polyglutamine diseases. The next challenge will be to fully understand the role of NLK in these diseases, which may aid future efforts to develop new treatments.

diseases, along with Huntington's disease, dentatorubral-pallidoluysian atrophy, and spinocerebellar ataxia (SCA) types 1, 2, 3, 6, 7, and 17. PolyQ expansion renders the host protein toxic, resulting in the formation of mutant protein aggregates and cell death; and the commonalities in the nature of the mutation and the presentation of the different polyQ disorders suggest the presence of a common pathogenic mechanism (*Orr, 2001*). Nonetheless, this mechanism has remained elusive and to date there are no cures or even effective therapies for most of these diseases.

AR is a well-studied steroid hormone receptor that also plays a crucial role in additional diseases including androgen insensitivity syndrome and prostate cancer (*Bennett et al., 2010*). Studies focusing on wild-type AR function and its role in other disease contexts can therefore shed light on SBMA pathogenesis. For instance, the main function of AR is to bind androgenic hormones, either testosterone or 5α-dihydrotestosterone (DHT), in the cytoplasm, and then translocate into the nucleus to act as a DNA-binding transcription factor that regulates androgen-dependent target gene expression (*Bennett et al., 2010*). SBMA pathogenesis is dependent upon the presence of circulating androgens and is therefore only observed in males, with homozygous female carriers showing only mild symptoms (*Katsuno et al., 2012*). The importance of androgens to the disease has also been clearly shown in mouse models of SBMA (*Katsuno et al., 2002*; *Chevalier-Larsen et al., 2004*). Furthermore, the nuclear translocation of AR is also crucial for pathogenesis (*Takeyama et al., 2002*; *Montie et al., 2009*; *Nedelsky et al., 2010*). It has also been suggested that an AR interdomain interaction known as the amino (N)-terminal–carboxy (C)-terminal (N/C) interaction is important for SBMA (*Orr et al., 2010*), as are the DNA-binding ability of AR (*Nedelsky et al., 2010*) and its post-translational modification including acetylation (*Montie et al., 2011*), methylation (*Scaramuzzino et al., 2015*), and other modifications (*Katsuno et al., 2012*). In addition, several cofactors and regulators of AR can influence SBMA disease pathogenesis (*McCampbell et al., 2000*; *Taylor et al., 2003*; *Palazzolo et al., 2007*; *Suzuki et al., 2009*; *Nedelsky et al., 2010*; *Montie et al., 2011*). Despite extensive studies, however, a precise molecular explanation for SBMA pathology has remained elusive.

Given the importance of androgens to SBMA pathogenesis, many approaches to SBMA therapeutics have focused upon depleting androgen levels in patients (*Banno et al., 2009*; *Katsuno et al., 2010b*; *Fernández-Rhodes et al., 2011*; *Yamamoto et al., 2013*). Unfortunately, these strategies have not yielded significant results in clinical trials; hence, new approaches are necessary. It has been shown that within prostate cancer cells, wild-type AR physically interacts with Nemo-like

kinase (NLK) and that NLK is able to regulate the activity and transcription of AR in this context (*Emami et al., 2009*). Interestingly, studies show that NLK interacts either directly or indirectly with a number of neurodegenerative disease-related proteins (*Lim et al., 2006*; *Ju et al., 2013*), suggesting that it may play an important role in the pathogenesis of neurodegenerative proteinopathies. Indeed, we have found that loss of one copy of *Nlk* (resulting in a 50% reduction in protein expression) is beneficial in mouse models of the polyQ disease SCA1 (*Ju et al., 2013*). NLK is an evolutionarily conserved mitogen-activated protein kinase -like serine/threonine kinase primarily studied in lower model organisms, where it has been linked to a number of signaling pathways (*Ishitani et al., 1999*; *Ohkawara et al., 2004*; *Ishitani et al., 2010*; *Ishitani and Ishitani, 2013*). In this study, we tested the hypothesis that NLK may play a role in SBMA pathogenesis. We present evidence that NLK influences the aggregation and toxicity of polyQ-expanded AR across multiple model systems, using cell culture, *Drosophila*, and mouse. Loss of one copy of *Nlk* was able to partially rescue disease phenotypes in both *Drosophila* and mouse models of SBMA. Furthermore, this 50% reduction in NLK protein expression dramatically extended the lifespan of SBMA mice. Finally, we investigated the molecular mechanisms by which NLK mediates these effects on SBMA and suggest a model in which NLK interacts with and phosphorylates AR, inhibiting its intramolecular N/C interaction and thereby promoting gene transcription via the AR activation function 2 (AF-2) domain. This effect on AR activity could then modulate SBMA-related aberrant AR-dependent gene transcription. In addition, reduced NLK expression can rescue the toxic effects of an N-terminal fragment of AR, suggesting that NLK can regulate the mutant AR protein—even in the absence of DNA binding and AR-responsive gene transcription.

## Results

### NLK interacts with the wild-type and mutant AR

It was previously reported that NLK could interact with the wild-type AR in prostate cancer cell lines (*Emami et al., 2009*). However, since SBMA is caused by polyQ-expanded AR (*La Spada et al., 1991*), and polyQ expansion can alter the ability of AR to interact with its binding partners (*Hsiao et al., 1999*; *Irvine et al., 2000*; *Sopher et al., 2004*), we tested if NLK could bind mutant AR. We co-transfected a FLAG-tagged wild-type NLK construct (FLAG-NLK-WT) with either wild-type or mutant HA-tagged human AR (HA-AR25Q and HA-AR120Q, respectively) into NSC-34 motor neuron-derived cells (*Cashman et al., 1992*) and performed co-immunoprecipitation (co-IP) assays. We found that NLK was able to co-IP both wild-type and mutant AR (*Figure 1A*). Interestingly, polyQ expansion led AR to be co-immunoprecipitated to a greater extent given its lower expression level (*Figure 1B*). Although future in vitro and in vivo experiments would be needed to verify this result, it was consistent in our hands. In addition, NLK was able to co-IP an N-terminal fragment of AR spanning the first 130 amino acids and containing the polyQ repeat, suggesting that NLK binds within this region (*Figure 1C*). It is worth mentioning that this fragment expresses as a doublet, and NLK seems to interact with only one of the forms of this fragment. We suspect that the upper band represents a post-translational modification of the fragment but further experiments would be required to confirm and expand this hypothesis.

### NLK enhances mutant AR aggregation in a kinase activity-dependent manner

We next wondered whether NLK could modulate SBMA disease phenotypes. PolyQ expansion results in the aggregation of the host protein, and inclusion formation is a pathological hallmark of polyQ and other neurodegenerative diseases (*Orr, 2001*; *Todd and Lim, 2013*). We therefore asked if NLK could influence the ability of the polyQ-expanded AR to aggregate. Mutant AR forms large polyQ- and DHT-dependent aggregates that can be readily visualized in our cell model via immunofluorescence (*Figure 1D* and *Figure 1—figure supplements 1–3*). Co-expression of wild-type NLK (NLK-WT) significantly increased the number of cells containing visible aggregates in DHT-treated, mutant AR (AR120Q)-expressing NSC-34 cells (*Figure 1E,G*), but did not cause a significant increase in aggregation in the absence of the AR ligand (*Figure 1G* and *Figure 1—figure supplement 1*). This increase was polyQ-dependent, as NLK co-expression resulted in only minimal aggregation in cells expressing a non-pathogenic AR25Q protein (*Figure 1G* and *Figure 1—figure supplements 2, 3*). Furthermore, this increase in aggregation was not detected when we used NLK-KN (*Figure 1F,G*), which harbors a lysine to methionine substitution at residue 155 and is defective for kinase activity (*Ishitani et al., 1999*). Importantly, co-expression of NLK does not alter the subcellular localization of non-aggregated AR or

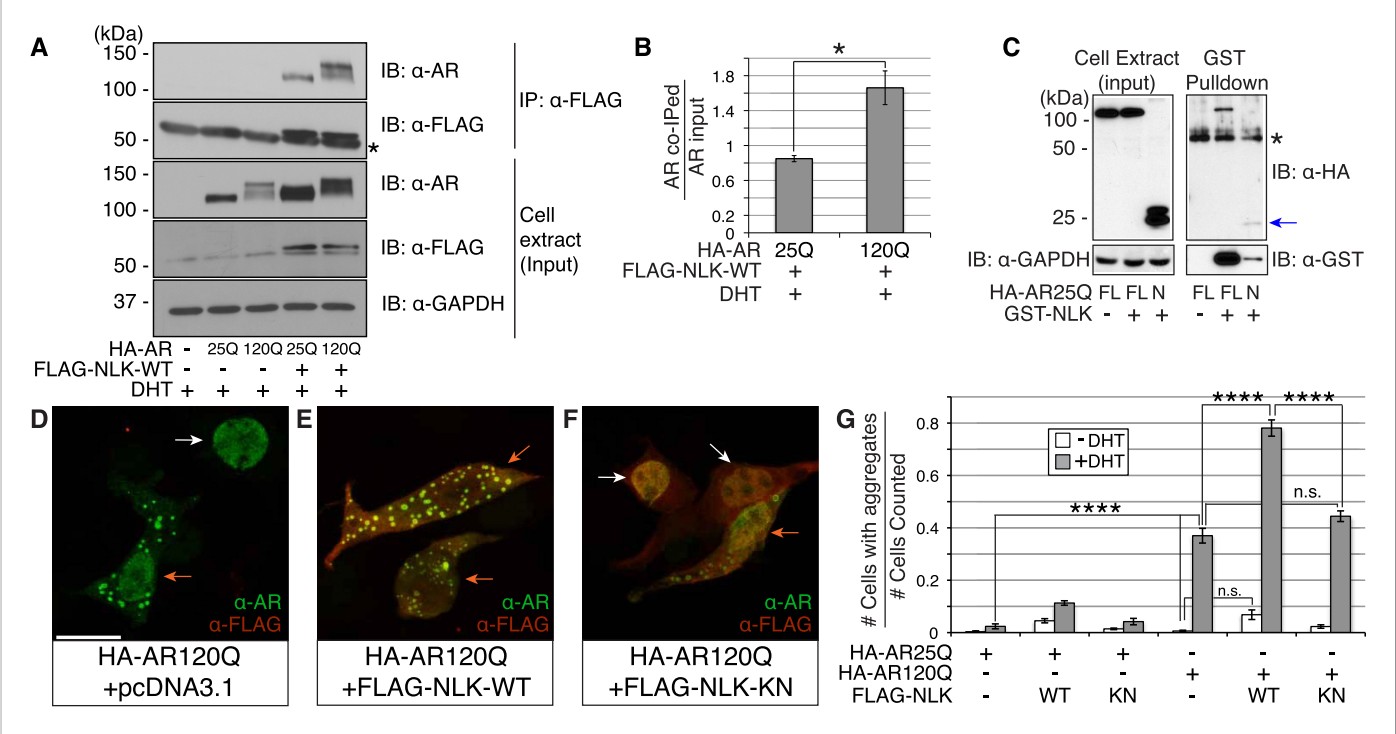

**Figure 1**. Nemo-like kinase (NLK) interacts with the mutant AR and enhances its aggregation. (**A**) NLK interacts with the AR protein in NSC-34 cells treated with 10 nM DHT. IP: immunoprecipitation. IB: immunoblot. GAPDH was used as a loading control in this and all following analyses unless otherwise specified. Asterisk marks a band corresponding to the immunoglobin heavy chain. (**B**) Quantification of co-IPed AR over total AR in input. *p < 0.05 (*t-test*). n = 3 trials. Error bars are standard error of the mean in this and all following graphs unless otherwise specified. (**C**) NLK interacts with the N-terminal region of AR. Both full-length (FL) and an N-terminal fragment (N, arrow) of AR were pulled down with NLK. Asterisk marks a non-specific band. (**D–G**) NLK enhances the formation of mutant AR aggregates in a kinase activity-dependent manner. NSC-34 cells were treated with DHT as indicated and subjected to immunofluorescence using anti-AR N-20 (green) and anti-FLAG (red) antibodies to detect AR aggregation and NLK co-expression, respectively. NLK-WT: wild-type NLK. NLK-KN: kinase-dead NLK. Representative images of DHT-treated cells are shown in (**D–F**). Images of the non-DHT-treated and AR25Q-expressing cells can be found in the *Figure 1—figure supplements 1–3*. Scale bar in (**D**) is 20 μm and refers to all three images. Cells were scored as containing aggregates (orange arrows) or not (white arrows) and the ratio of aggregate-positive cells out of total scored is quantified in (**G**). n.s. = not significant, ****p < 0.0001 (*ANOVA* with Tukey's *post-hoc* analysis). n ≥ 3 trials.

The following figure supplements are available for figure 1:

**Figure supplement 1**. PolyQ-expanded AR120Q does not aggregate in the absence of DHT.

**Figure supplement 2**. Non-pathogenic AR25Q shows diffuse cytoplasmic localization in the absence of DHT.

**Figure supplement 3**. Non-pathogenic AR25Q undergoes nuclear translocation in response to DHT, but largely does not aggregate.

**Figure supplement 4**. Mutant AR forms high molecular weight aggregates in the stacking gel of SDS–PAGE gels.

inhibit its nuclear translocation, although cells with robust aggregation often showed a slight reduction in nuclear staining, suggesting that much of the mutant protein was sequestered into aggregates in these cells. In addition, we did not recognize any obvious changes in subcellular localization between NLK-WT and NLK-KN, which could both be detected in the cytoplasm and nucleus. We also noticed that aggregated mutant AR protein could be detected biochemically in the stacking gel when we ran DHT-treated NSC-34 cell extracts on SDS-PAGE gels. Co-expression of NLK-WT increases this aggregation (*Figure 1—figure supplement 4*). Taken together, these data suggest that NLK is able to affect polyQ-AR-specific defects within this cell culture system in a kinase activity-dependent manner.

## NLK increases mutant AR aggregation in primary spinal cord motor neurons

To test whether NLK can also increase mutant AR aggregation in an in vivo motor neuron setting, we cultured mouse primary motor neurons from spinal cord and transfected with GFP-tagged polyQ-expanded mutant AR and either a control plasmid or FLAG-tagged NLK-WT in the presence or absence of DHT. Cells were then blindly scored for the presence or absence of aggregation. We found that NLK is able to robustly increase mutant AR aggregation in DHT-treated neurons, while it only modestly increased aggregation in the absence of hormone (*Figure 2*).

## NLK modulates mutant AR toxicity in a *Drosophila* model of SBMA

Having established that NLK can modulate the aggregation of the mutant AR in our cell culture system, we went on to determine the effect of modulating NLK activity and expression on SBMA in model organisms. We began by utilizing *Drosophila*. When a full-length *AR* transgene is expressed in the *Drosophila* eye via the Gal4/UAS system (*Brand and Perrimon, 1993*), it produces a polyQ-, DHT-dependent retinal degeneration phenotype characterized by the presence of fused ommatidia and abnormal interommatidial bristles along the posterior margin of the eye (*Figure 3A,B* and *Figure 3—figure supplement 1*). This phenotype is similar to what has been reported for other full-length mutant AR *Drosophila* models of SBMA (*Takeyama et al., 2002*; *Pandey et al., 2007*; *Nedelsky et al., 2010*). We crossed SBMA flies to flies that were heterozygous for a loss-of-function mutation in the fly homolog of *Nlk*, *nemo* (*nmo*). To ensure that this was not due to a non-specific background effect, we utilized two independent *nmo* loss-of-function alleles (adk1 and adk2) (*Verheyen et al., 2001*). Both alleles were able to partially, but consistently, suppress the mutant AR-mediated rough eye phenotypes (*Figure 3C,D*), although the $nmo^{adk2}$ line showed a more profound rescue than $nmo^{adk1}$. Next, we assessed whether this phenotype correlated with a change in mutant AR aggregation. To do this, we compared protein extracts from *Drosophila* heads of each genotype by immunoblot. Aggregation of the mutant AR protein can be detected as a smear in the stacking gel and was increased in flies raised in the presence of DHT (*Figure 3E,F*). Loss of one copy of *nmo* tended to reduce this aggregation, particularly when assessed with the $nmo^{adk2}$ allele (*Figure 3E,F*). Although this reduction in aggregation failed to reach significance by *ANOVA,* the difference seen between the two alleles correlates with the more profound partial rescue of the mutant AR-dependent retinal degeneration seen with $nmo^{adk2}$ compared to $nmo^{adk1}$ in this fly model of SBMA.

We next tested whether increased expression of NLK could enhance the mutant AR phenotypes in this *Drosophila* SBMA model. To do this, we crossed SBMA flies with flies expressing either the human NLK or an EGFP control (*Figure 4*). Co-expression of NLK-WT enhanced the retinal degeneration phenotype (*Figure 4B*) and, more strikingly, dramatically increased the mutant AR aggregation detected by immunoblot (*Figure 4D*, lane 4 vs lane 6). Once again, this phenotype was DHT dependent (*Figure 4D*, lane 5 vs lane 6). Importantly, we also found that expression of kinase-dead NLK-KN did not enhance the retinal degeneration phenotype (*Figure 4C*) or mutant AR aggregation (*Figure 4D*, lane 4 vs lane 7), a finding consistent with our cell culture data (*Figure 1*). Taken together, these studies strongly suggest that NLK exacerbates the toxicity of the polyQ-expanded mutant AR via a mechanism that depends upon its kinase activity.

## Decreased NLK expression improves disease pathology in a SBMA mouse model

Our cell culture and *Drosophila* data strongly suggest that reducing NLK expression or activity will be beneficial in SBMA, but we wished to confirm this at the mammalian level. We therefore decided to make use of our previously produced *Nlk* mutant mice (*Ju et al., 2013*). Mice heterozygous for either of two gene trap alleles (both simply referred to as $Nlk^{gt/+}$ here) show a 50% reduction in NLK expression, while mice homozygous for the gene trap alleles show an approximately 90% reduction in protein expression (*Ju et al., 2013*). Importantly, this decrease can be detected in both the spinal cord and skeletal muscle (*Figure 5*), the two tissues primarily affected in SBMA. We also obtained mice that express a BAC transgene containing a 121Q *AR* and its endogenous regulatory elements (*BAC fxAR121*). These mice recapitulate key SBMA disease phenotypes, including motor neuron pathology, muscle atrophy, and early lethality. These phenotypes are only seen in male mice, as is consistent with the hormone specificity of this disease (*Cortes et al., 2014*). As homozygous expression

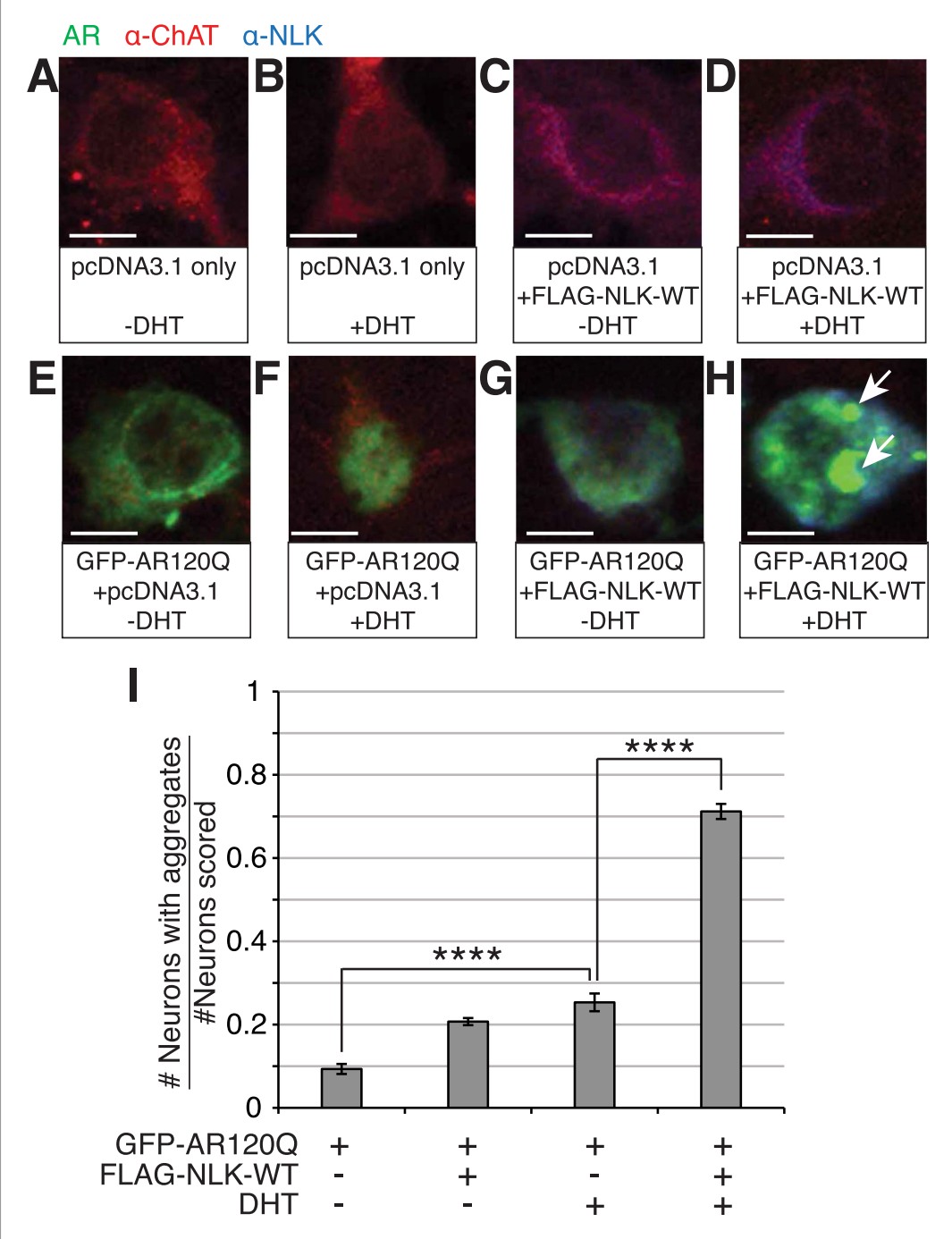

**Figure 2**. NLK increases mutant AR aggregation in primary motor neurons. (**A–H**) Primary motor neurons were transfected with GFP-tagged AR120Q, FLAG-tagged NLK-WT, or a pcDNA3.1 empty vector control and treated with 10 µM DHT. Aggregation was analyzed by immunofluorescence at 9 days in vitro (DIV). An antibody to choline acetyltransferase (ChAT) was used to confirm motor neuron identity and is shown in red. GFP-AR120Q is shown in green and NLK co-expression (as detected by an NLK antibody) is in blue. All images were collected using identical confocal settings. In the absence of DHT, AR localizes to the cytoplasm (**E**, **G**), while DHT induces its nuclear translocation (**F**) and its aggregation, which is enhanced by NLK (**H**). Arrows mark aggregates, which can be detected in both the nucleus and cytoplasm. Scale bars are 10 µm. (**I**) The number of neurons containing aggregates out of total scored was quantified and averaged over different regions of the plate. At least 140 neurons were scored per condition. ****p < 0.0001 (*ANOVA* with Tukey's *post-hoc* analysis).

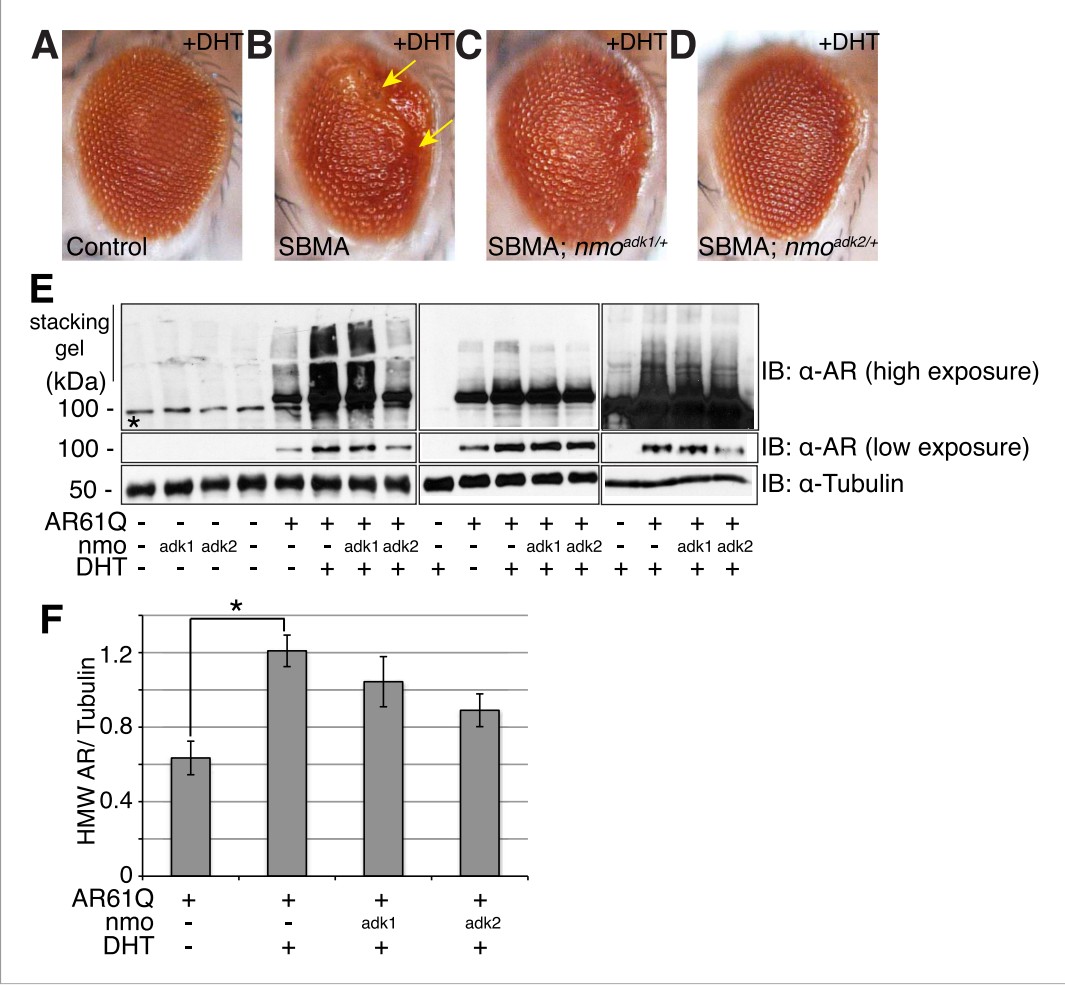

**Figure 3**. NLK genetically interacts with the mutant AR in *Drosophila*. Loss of one *nmo* allele suppresses mutant AR-mediated SBMA phenotypes in *Drosophila*. (**A–D**) Light microscopy of adult *Drosophila* eyes is shown. In (**B**), arrows mark a DHT-dependent retinal degeneration phenotype along the posterior margin. Flies were raised at 30°C and genotypes are as follows: (**A**) *GMR-Gal4/+; UAS-EGFP/+*, (**B**) *GMR-Gal4, UAS-AR61Q/+*, (**C**) *GMR-Gal4, UAS-AR61Q/+; nmo^adk1/+^*, (**D**) *GMR-Gal4, UAS-AR61Q/+; nmo^adk2/+^*. For all panels, experiments were repeated multiple times and representative images are shown. (**E**) Western blots from three different trials show the aggregation of the mutant AR as a smear in the stacking gel at high exposure. Lower exposure reveals the AR61Q monomer at the expected size of around 110 kDa. Asterisk marks a non-specific band present in all lanes. (**F**) High molecular weight (HMW) or aggregated AR was quantified as compared to the tubulin loading control and averaged over trials. *$p <$ 0.05 (*ANOVA* with Tukey's *post-hoc* analysis). $n \geq 3$ trials.

The following figure supplement is available for figure 3:

**Figure supplement 1**. Expression of a full-length AR protein in the *Drosophila* eye results in polyQ- and DHT-dependent retinal degeneration phenotypes.

of the *Nlk* gene trap alleles is lethal, we carried out our analysis in the heterozygous background. *Nlk^gt^* heterozygous mice were crossed to *BAC fxAR121* mice and their F1 male progeny were analyzed to determine if loss of one copy of *Nlk* could rescue the SBMA-related phenotypes seen in the *BAC fxAR121* mice. For our analysis, we began by looking at motor neuron pathology. *BAC fxAR121* mice, like other SBMA mouse models (*Chevalier-Larsen et al., 2004*; *Yu et al., 2006*), fail to show overt motor neuronal loss. There is, however, a pathogenic decrease in the area and perimeter of the spinal motor neuron soma in this model (*Cortes et al., 2014*). We analyzed L4–L5 anterior horn motor neurons and found that a reduction in NLK expression resulted in significantly larger motor neuron cell bodies

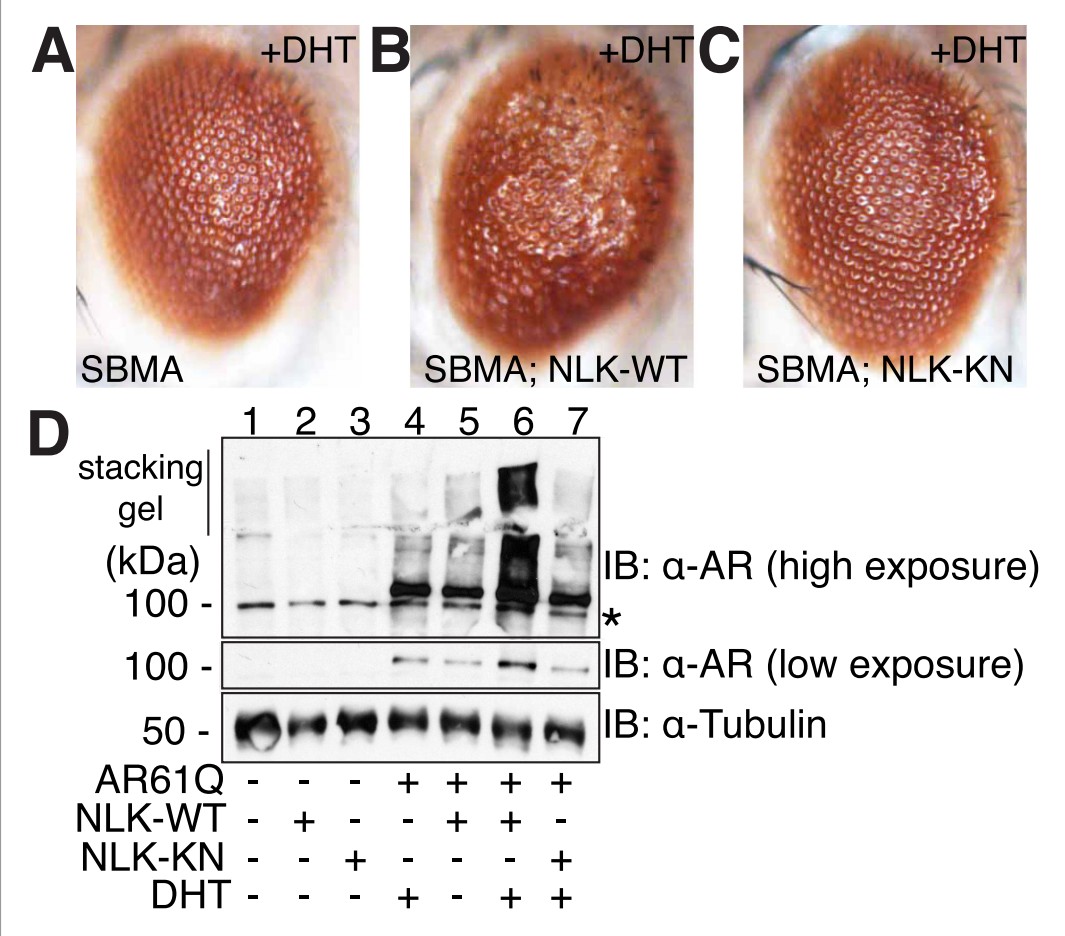

**Figure 4**. NLK modulates mutant AR phenotypes in *Drosophila* in a kinase activity-dependent manner. (**A**–**C**) Light microscopy of adult *Drosophila* eyes is shown. Flies were raised at 30°C and genotypes are as follows: (**A**) *GMR-Gal4, UAS-AR61Q/UAS-EGFP*, (**B**) *GMR-Gal4, UAS-AR61Q/UAS-NLK-WT*, (**C**) *GMR-Gal4, UAS-AR61Q/UAS-NLK-KN*. (**D**) Mutant protein aggregation is shown by immunoblot with indicated genotypes. Aggregated mutant AR protein can be detected as a smear in the stacking gel at higher exposures, while the AR61Q monomer expresses at around 110 kDa and can be seen at lower exposures. Asterisk marks a non-specific band present in all lanes. For all panels, experiments were repeated multiple times, and representative images are shown.

than those seen in SBMA littermates (*Figure 6*), suggesting an improvement in pathology. We next focused on muscle pathology, since muscle cramping and atrophy are prominent symptoms in SBMA patients (*Rhodes et al., 2009*; *Katsuno et al., 2012*), and this SBMA mouse model shows an obvious muscle atrophy phenotype (*Cortes et al., 2014*; *Lieberman et al., 2014*). Compared to wild-type and *Nlk^{gt/+}* mice, *BAC fxAR121^{+/−}* mice showed a reduction in the Feret's diameter and cross-sectional area of muscle fibers, as well as more angulated fibers and increased connective tissue, all of which is suggestive of atrophy (*Figure 7A–E* and *Figure 7—figure supplement 1*). Although muscle atrophy phenotypes were still apparent in *BAC fxAR121^{+/−}; Nlk^{gt/+}* mice, the average fiber size was significantly increased compared to their *BAC fxAR121^{+/−}* littermates (*Figure 7C–E*). This increase was apparent at 20 weeks (mid-late symptomatic stage) and 30 weeks (late symptomatic stage) of age, but was not seen at disease onset at 10 weeks of age and was no longer significant at very late disease stages at 40 weeks of age (*Figure 7E* and *Figure 7—figure supplement 1*).

We also stained muscles for NADH transferase activity (*Figure 7F–I*), as defects in the patterning of this stain are seen in SBMA mouse models and are indicative of pathology (*Sopher et al., 2004*; *Monks et al., 2007*; *Palazzolo et al., 2009*). As previously reported (*Cortes et al., 2014*), there was a general increase in staining in the muscle of *BAC fxAR121^{+/−}* mice (*Figure 7H*), as opposed to the

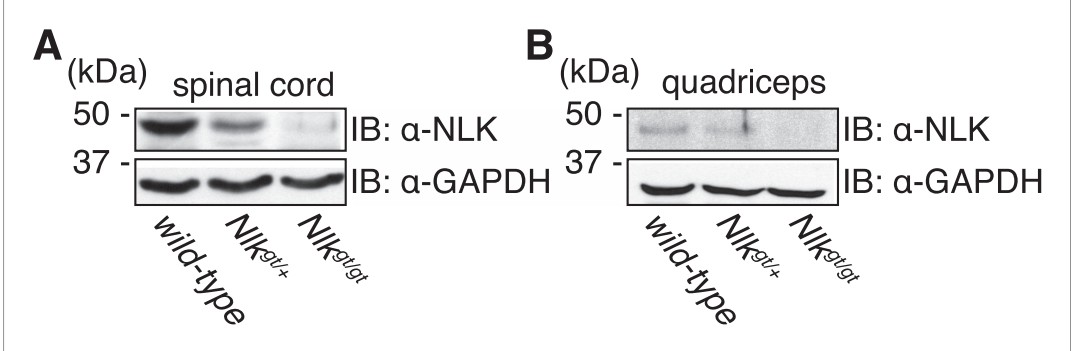

**Figure 5**. *Nlk^gt* mice show reduced NLK expression in the spinal cord and skeletal muscle. Whole spinal cord (**A**) and quadriceps (**B**) extracts from indicated genotypes were immunoblotted with a NLK antibody. Mice heterozygous for *Nlk^gt* show a 50% reduction in protein expression, while mice homozygous for the allele show an approximately 90% reduction. GAPDH was used as a loading control.

normal 'checkerboard' pattern seen in wild-type and *Nlk^gt/+* mouse muscle (*Figure 7F,G*). Consistent with the increase in fiber size, *BAC fxAR121^+/−*; *Nlk^gt/+* mice also showed a partial but consistent rescue in this phenotype at 30 weeks of age compared to their littermate controls (*Figure 7I*). We quantified this change in staining intensity by measuring the mean gray value of the images (*Figure 7—figure supplement 2*).

## Loss of one copy of *Nlk* extends the lifespan of a SBMA mouse model

*BAC fxAR121* mice show an early lethality phenotype that can be completely rescued by removing the mutant AR only from the skeletal muscle (*Cortes et al., 2014*; *Lieberman et al., 2014*). This early

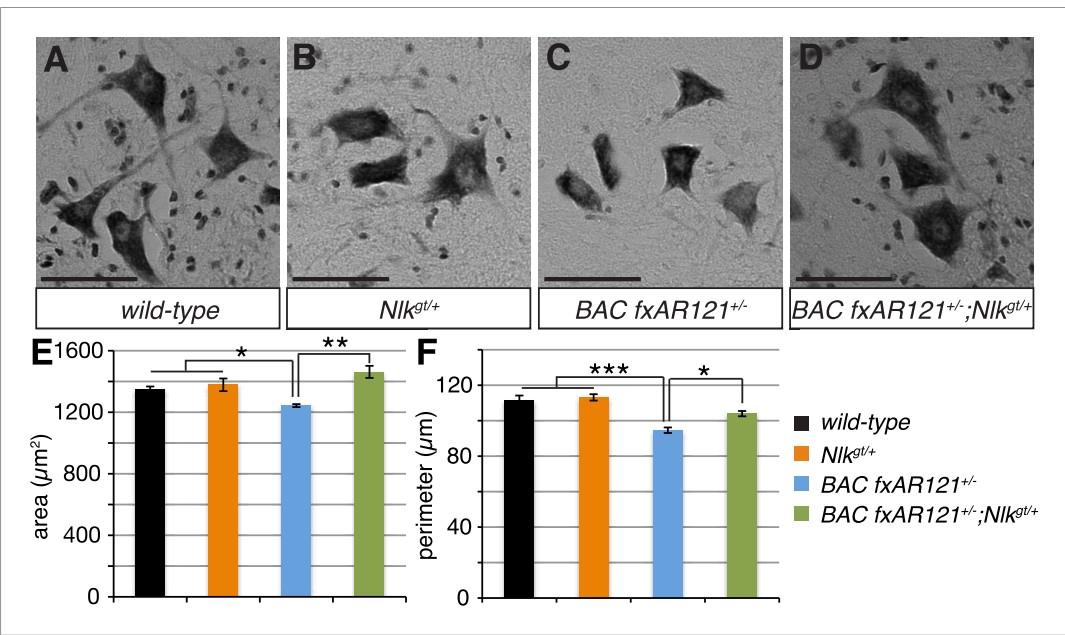

**Figure 6**. Loss of one copy of *Nlk* improves the pathogenic change in motor neuronal soma size in SBMA mice. (**A–D**) Spinal cord cross-sections from the L4–L5 region were stained with cresyl violet (nissl stain) to visualize the spinal motor neuron cell bodies. Representative images from the anterior horn region of 40-week-old mice are shown. Scale bars are 50 μm. (**E, F**) The average motor neuron area (**E**) and perimeter (**F**) were measured and averaged over genotype. n = 2, 4, 4, 3 per genotype, respectively. Over 100 neurons were scored per animal. *p < 0.05, **p < 0.01, ***p < 0.001 (*ANOVA* with Tukey's *post-hoc* analysis).

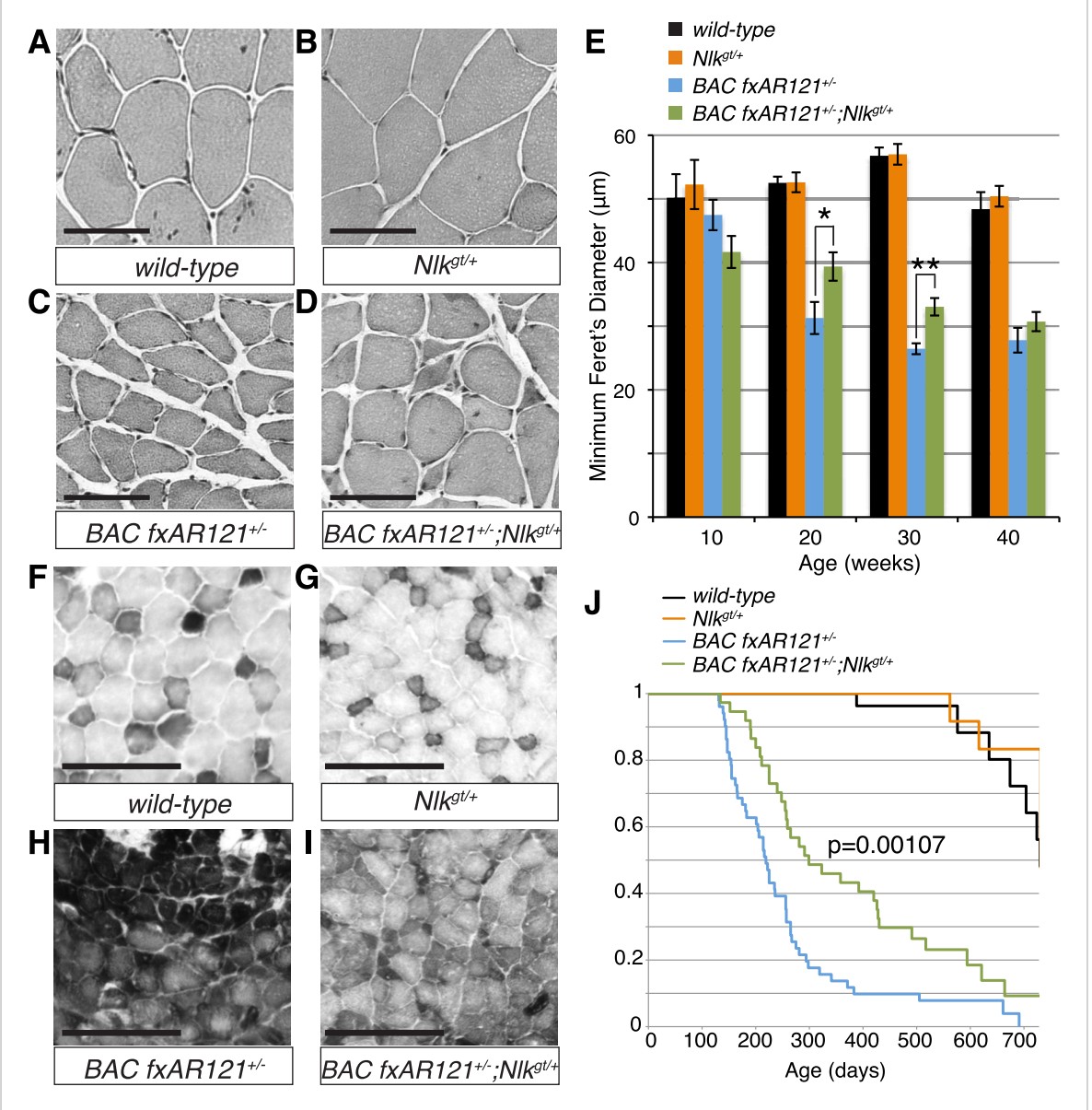

Figure 7. Loss of one copy of *Nlk* significantly rescues SBMA phenotypes in mice. (**A**–**D**) Mouse quadriceps sections of indicated genotypes were stained for hematoxylin and eosin and representative 30-week-old images are shown. Scale bars are 50 µm. (**E**) Quantification of the average minimum Feret's diameter of muscle fibers at ages indicated. *p < 0.05, **p < 0.005 (*t-test*). For 10 weeks, *n* = 3, 3, 4, and 3 per genotype, respectively. For 20 weeks, *n* = 4, 3, 5, and 5. For 30 weeks, *n* = 7, 5, 5, and 8. For 40 weeks, *n* = 2, 5, 4, and 3. More than 500 fibers were scored per animal. See also *Figure 7—figure supplement 1*. (**F**–**I**) Reduced NLK expression improves the defective NADH transferase activity pattern seen in *BAC fxAR121+/−* mouse muscle. Six littermate sets were compared and representative images at 30 weeks of age are shown. Scale bars are 200 µm. See also *Figure 7—figure supplement 2*. (**J**) Kaplan–Meier survival analysis shows a significant extension in the lifespan of *BAC fxAR121+/−* mice with a 50% reduction of NLK. p = 0.00107 (*log rank test*). n = 27, 27, 51, and 37 per genotype, respectively.

The following figure supplements are available for figure 7:

**Figure supplement 1**. Loss of one copy of *Nlk* increases muscle fiber size in *BAC fxAR121+/−* mouse quadriceps.

**Figure supplement 2**. A 50% reduction in NLK expression reduces aberrant NADH transferase staining in 30-week-old SBMA mice.

lethality can be recapitulated in our C57/129 hybrid genetic background (*Figure 7J*; median survival of 219 days), although the mice live slightly longer than on the pure C57BL/6J background. Strikingly, in addition to rescuing the muscle atrophy and motor neuron phenotypes, loss of one copy of *Nlk* extended the lifespan of the *BAC fxAR121*$^{+/-}$ mice (*Figure 7J*; increased to a median survival of 299 days). This effect is dramatic considering that these mice have only a 50% reduction in NLK protein expression. Loss of one copy of *Nlk* alone did not significantly alter lifespan (*Figure 7J*, orange vs black lines, p = 0.854, log rank test).

## Loss of one copy of *Nlk* decreases mutant protein aggregation in a SBMA mouse model

As NLK influences the aggregation of the polyQ-expanded AR in cell culture and *Drosophila* (*Figures 1–4*), we tested if there was any change in the aggregation of mutant AR in the *BAC fxAR121* mice when NLK expression was decreased. While the mutant AR shows primarily diffuse staining in the spinal motor neuron nuclei (data not shown), we were able to detect aggregates in the skeletal muscle of *BAC fxAR121* mice via multiple assays. First, aggregation could be detected via immunofluorescence with AR antibodies, resulting in punctate, nuclear staining that was absent from wild-type or *Nlk*$^{gt/+}$ muscle (*Figure 8A–D*). Loss of one copy of *Nlk* significantly reduced the number of nuclei containing aggregates by 20 weeks of age (*Figure 8E*). Next, we analyzed aggregation biochemically. When muscle protein extracts were subjected to a filter trap assay, insoluble, aggregated AR was detected specifically in *BAC fxAR121*$^{+/-}$ and *BAC fxAR121*$^{+/-}$; *Nlk*$^{gt/+}$ samples, and not in wild-type or *Nlk*$^{gt/+}$ samples (*Figure 8F*). Quantification revealed that the amount of aggregated AR was significantly decreased with loss of one copy of *Nlk* by 20 weeks of age, although there was no longer a difference in this phenotype at very late stages of disease (i.e., 40 weeks) (*Figure 8G*). At late stages of the disease, the mutant AR could also be detected as a high molecular weight smear in the stacking gel of SDS-PAGE gels, and, once again, this was decreased with loss of one copy of *Nlk* (*Figure 8H,I*). Therefore, as was seen in cell culture, primary motor neurons, and flies, NLK promotes the aggregation of mutant AR, and this aggregation positively correlates with an exacerbation of SBMA phenotypes. Conversely, loss of one copy of *Nlk* reduces aggregation across models, and we have found that this 50% reduction in NLK protein is sufficient to significantly improve SBMA-related phenotypes, including lifespan, in *BAC fxAR121* SBMA mice.

## NLK induces the phosphorylation of AR

Having established that NLK promotes SBMA phenotypes, we next wondered what was the molecular mechanism underlying this effect. Since NLK binds AR (*Figure 1A–C*) and is a kinase, we first tested whether NLK could phosphorylate AR. We noted that co-expression of AR with NLK-WT induced an electrophoretic mobility shift in the AR protein (*Figure 9A*, lane 2, blue arrow) that was not seen with co-expression of NLK-KN (*Figure 9A*, lane 3). This mobility shift was reversed when cell extracts were incubated with lambda phosphatase (*Figure 9A*, lane 5, red arrow), suggesting that this shift represents an NLK-induced AR phosphorylation. NLK targets proline-directed serines and threonines. There are thus seven potential NLK target sites within the full-length AR protein. We were able to obtain phospho-specific antibodies for two of these sites, serine (S)81 and S308. NLK significantly increased AR phosphorylation at both of these sites in a kinase activity-dependent manner (*Figure 9B–D*). Evidence suggests that NLK can target AR in both the presence and absence of hormone (data not shown), but as the effect of NLK on the non-ligand-bound AR is unlikely to be disease relevant, we have focused on its ligand-dependent activity. Taken together, NLK was able to interact with and regulate the phosphorylation of both the wild-type (data not shown) and polyQ-expanded AR at two sites, although NLK can likely target other sites in AR as well.

We next asked whether NLK could influence the phosphorylation of AR in vivo. We used the same phospho-specific antibodies to assess the phosphorylation of the mutant AR protein in the skeletal muscle of *BAC fxAR121* mice. Unfortunately, the phospho-AR-S308 antibody could not detect the mutant AR protein in these mice (data not shown), and so we could not assess if NLK influences phosphorylation at this site in vivo via this approach. However, the phospho-AR-S81 antibody could detect the mutant AR protein, and we found that male mice lacking one copy of *Nlk* showed a reduction in the level of AR-S81 phosphorylation (*Figure 9E,F*). This suggests that NLK regulates the phosphorylation of AR in vivo.

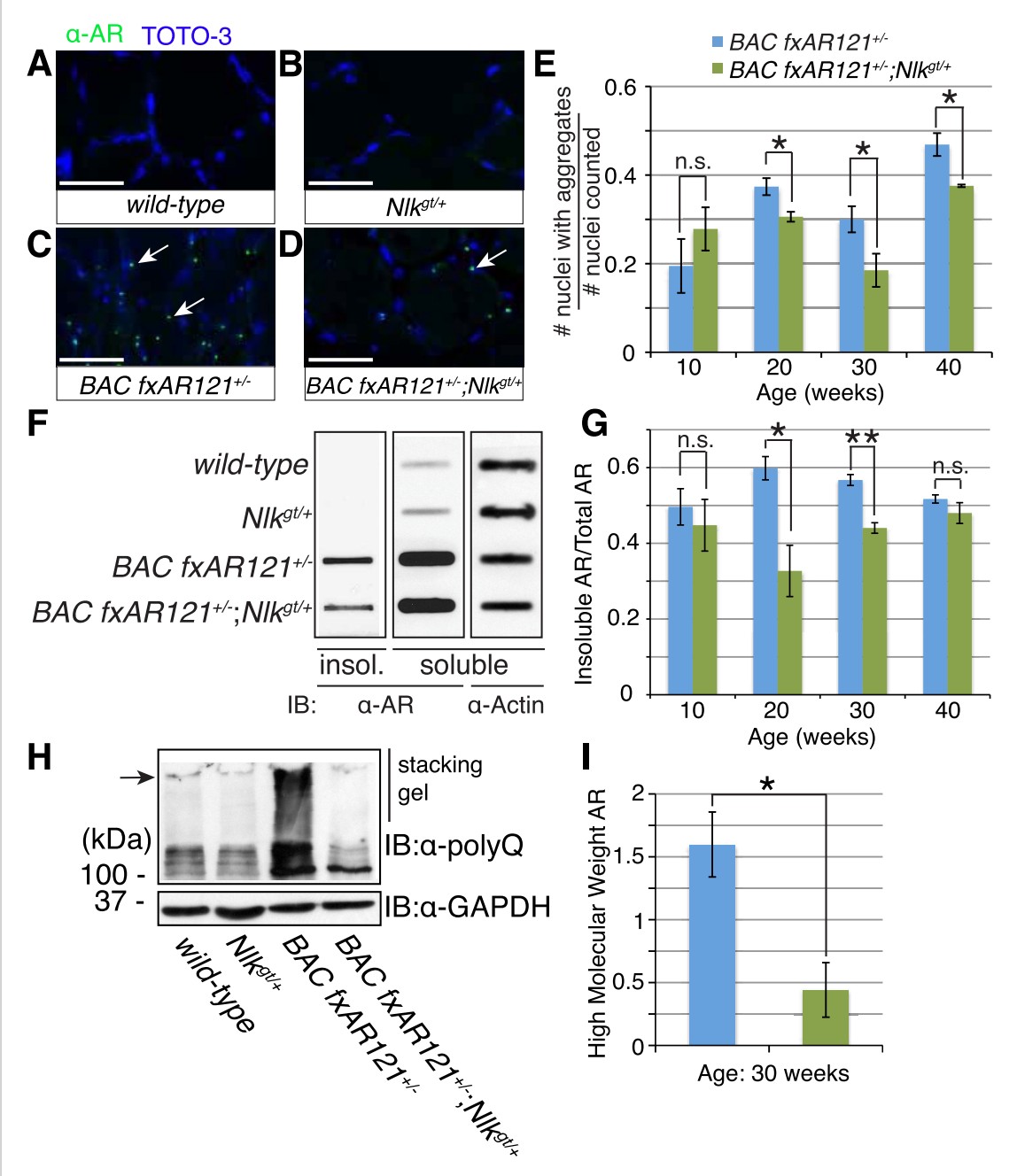

**Figure 8**. Loss of one copy of *Nlk* decreases mutant AR aggregation in mice. (**A–D**) Nuclear AR aggregates (arrows) can be detected in quadriceps of mice expressing the *BAC fxAR121* transgene (**C**, **D**), but not in controls (**A**, **B**). Representative 30-week-old samples are shown. Scale bars are 50 µm. Nuclei are marked with TOTO-3 in blue. (**E**) Quantification of the ratio of nuclei containing aggregates out of total nuclei counted; 300 to 500 fibers per mouse. n.s. = not significant, $*p < 0.05$ (*t-test*). For 10 weeks, $n = 3$ each. For 20 weeks, $n = 5$ each. For 30 weeks, $n = 5$ and 8, respectively. For 40 weeks, $n = 4$ and 3, respectively. (**F**) A representative filter trap assay blot from 20-week-old quadriceps samples. (**G**) The amount of insoluble (Insol.) AR out of total (Insol. + Soluble) was quantified. n.s. = not significant, $*p < 0.05$, $**p < 0.005$ (*t-test*). For 10 weeks, $n = 3$ each. For 20 weeks, $n = 5$ each. For 30 weeks, $n = 4$ each. For 40 weeks, $n = 4$ and 3, respectively. (**H**) A representative blot shows mutant AR retained in the stacking gel of SDS-PAGE gels as high molecular weight aggregates (arrow). 30-week-old quadriceps samples are shown. An antibody to the polyQ region (1C2) was used. (**I**) Quantification of AR in the stacking gel normalized to loading control. $*p < 0.05$ (*t-test*). $n = 3$ for each genotype.

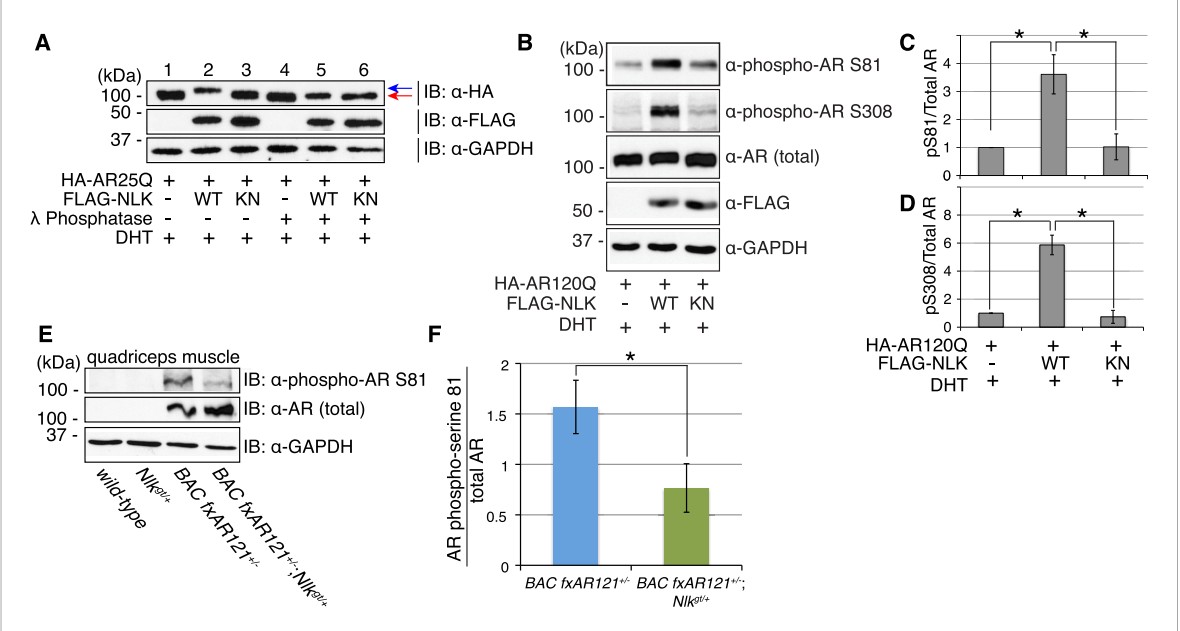

**Figure 9.** NLK influences the phosphorylation status of AR. (**A**) NLK can induce the phosphorylation of AR in a cell culture system. AR25Q is shown here, but the same effect is seen with polyQ-expanded AR. (**B–D**) NLK can phosphorylate the mutant AR at S81 and S308. (**C**) Quantification of phospho-AR-S81 expression over total AR expression (as detected by AR-N20 antibody). (**D**) Quantification of phospho-AR-S308 expression over total AR expression. *p < 0.05 (*t-test*). n ≥ 4 trials. (**E**, **F**) NLK can affect mutant AR phosphorylation in SBMA mouse muscle in vivo. (**E**) Representative image of 30-week-old mouse quadriceps samples immunoblotted with phospho-AR-S81 antibody and an antibody to detect total AR. Only mutant AR protein is shown here, but a lower wild-type AR band can also be detected in all 4 genotypes. (**F**) Quantification of phospho-AR-S81 expression over total AR expression. *p < 0.05 (*t-test*). n = 7 and 9 for *BAC fxAR121⁺/⁻* and *BAC fxAR121⁺/⁻; Nlk^{gt/+}*, respectively.

## NLK regulates the aggregation and toxicity of the mutant AR via phosphorylation

We next tested if the NLK-mediated change in AR-S81 phosphorylation contributed to the SBMA phenotype. As NLK increases mutant AR aggregation across multiple model systems (cultured cells, primary motor neurons, *Drosophila*, and mouse), and this positively correlates with its effects on SBMA phenotypes in vivo, we reasoned that our cell culture system could be reliably used as an initial read-out for NLK-mediated effects on mutant AR. In order to test the specific contribution of AR-S81 phosphorylation to SBMA-related phenotypes, we introduced a phospho-resistant mutation into the polyQ-expanded *AR* construct at S81 (S81A; serine to alanine substitution). We found that the AR-S81A mutant tended to show slightly less aggregation than AR-S81 (*Figure 10A*, representative images in *Figure 10—figure supplement 1*), although this decrease was not significant by *ANOVA*. Interestingly, the S81A mutation significantly compromised the NLK effect on mutant AR aggregation (*Figure 10A* and *Figure 10—figure supplement 1*). This suggests that phosphorylation at S81 can contribute to the NLK-mediated effects on AR aggregation at least in our cell culture system.

To further investigate the contribution of NLK-mediated AR-S81 phosphorylation on mutant AR toxicity, we decided to make use of a previously published N-terminal fragment model of SBMA. Expression of a polyQ-expanded 130 amino acid N-terminal fragment of AR (trAR¹¹²Q) in the *Drosophila* eye results in a robust retinal degeneration and depigmentation phenotype (*Chan et al., 2002*). This fragment is able to interact with NLK (*Figure 1C*) and contains only two putative NLK targets sites, S81 and S94. We found that mutating S94 to alanine did not affect the NLK-mediated increase in full-length mutant AR aggregation in NSC-34 cells (data not shown). We therefore predicted that if loss of one *Nlk* allele could rescue the toxicity of this polyQ-expanded AR N-terminal fragment, the mechanism would likely depend upon the interaction of NLK with AR and phosphorylation at S81. We first confirmed that NLK could still induce phosphorylation at S81 in this fragment by co-expressing the proteins in NSC-34 cells (*Figure 10B*). We next crossed trAR¹¹²Q flies with *nmo* mutant flies and

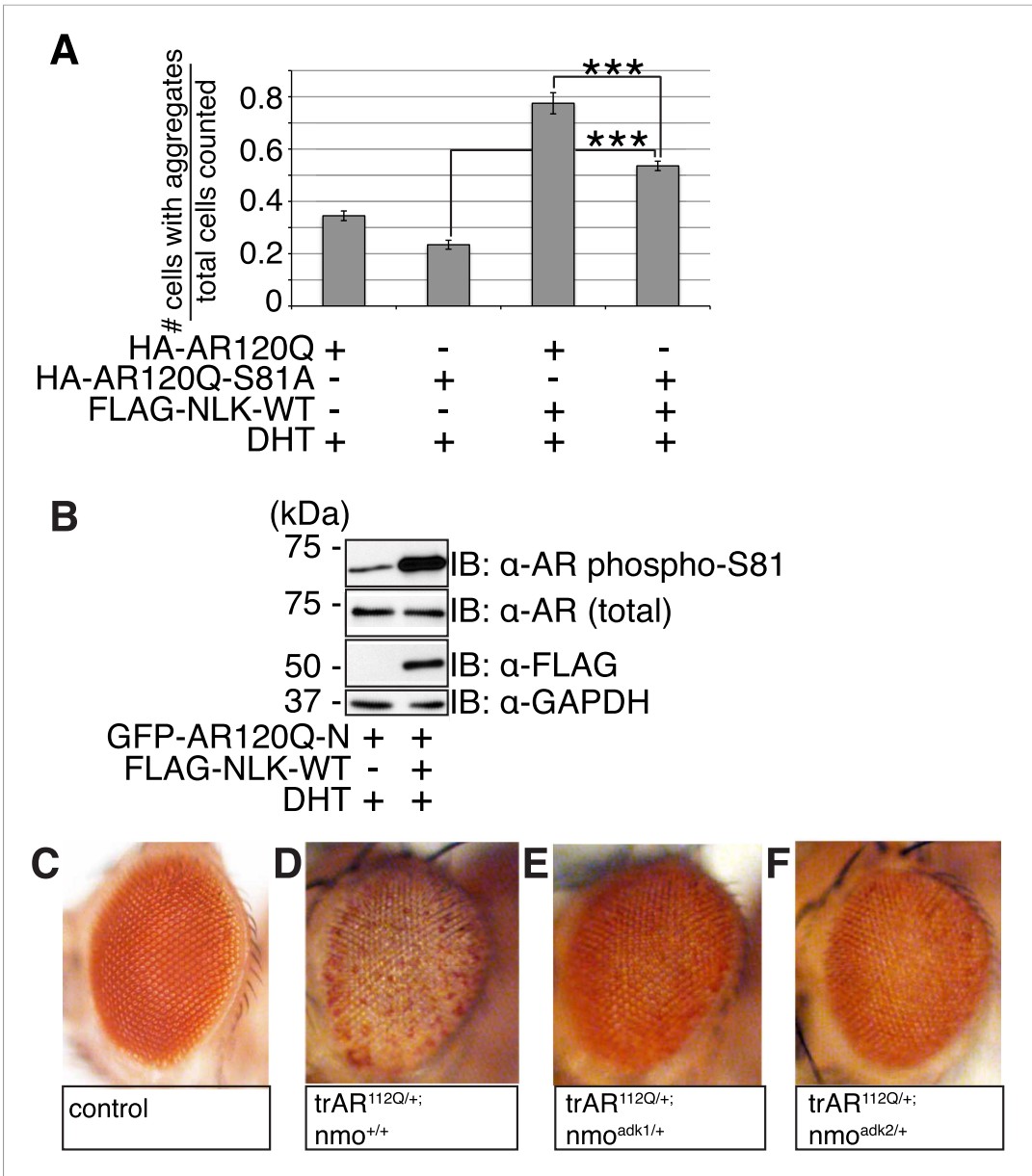

**Figure 10**. NLK regulates the aggregation and toxicity of mutant AR by influencing the phosphorylation of AR at residues including S81. (**A**) AR-S81 phosphorylation could contribute to the NLK effect on mutant AR aggregation. NSC-34 cells were transfected with indicated constructs and treated with 10 nM DHT. Quantification of the ratio of cells containing AR aggregates out of total counted is shown. ***p < 0.001 (*ANOVA* with Tukey's *post-hoc* analysis). n ≥ 3 trials. See also *Figure 10—figure supplement 1*. (**B**) NLK induces the phosphorylation of a 130 amino acid AR N-terminal fragment at S81 in NSC-34 cells. (**C–F**) Reduced expression of NLK suppresses the toxicity induced by a mutant AR fragment in a Drosophila model of SBMA. Two independent mutant alleles (adk1 and adk2) of *nmo* showed the same results. Flies were raised at 22°C and genotypes are as follows: (**C**) *GMR-Gal4/+; UAS-EGFP/+*, (**D**) *GMR-Gal4/+; UAS-trAR[112Q]/+*, (**E**) *GMR-Gal4/+; UAS-trAR[112Q]/nmo[adk1]*, (**F**) *GMR-Gal4/+; UAS-trAR[112Q]/nmo[adk2]*. More than 50 adult flies per genotype were observed at day 2 after eclosion, and five independent experiments were performed.

The following figure supplement is available for figure 10:

**Figure supplement 1**. S81 phosphorylation contributes to NLK-mediated effects on AR aggregation.

assessed the eye phenotypes of the resulting progeny. We found that loss of one copy of *nmo* reversed the depigmentation phenotype induced by the trAR[112Q] fragment (*Figure 10C–F*). This result supports the idea that NLK may regulate the aggregation and toxicity of polyQ-expanded AR via N-terminal binding and AR-S81 phosphorylation in SBMA.

## NLK promotes AR transactivation activity

SBMA is caused by polyQ expansion in the full-length AR protein, but the exact molecular mechanisms underlying the disease are unclear. On one hand, it has been reported that mutant AR can be processed by proteases and the polyQ-containing AR fragments are toxic and aggregation-prone (*Merry et al., 1998*; *Ellerby et al., 1999*; *Chan et al., 2002*). Mutant AR inclusions in patient tissue can only be detected by N-terminal AR antibodies and not by antibodies to the AR C-terminus (*Li et al., 1998*). It has therefore been speculated that aggregates are comprised of mostly N-terminal AR fragments, and there is some evidence from mouse models to support this theory (*Li et al., 2007*). These fragments lack the AR DNA-binding domain, suggesting that the polyQ-dependent toxicity seen, for example, in the trAR[112Q] *Drosophila* model must occur in the absence of specific DNA binding and AR-mediated gene transcription. Our data show that NLK can modulate mutant AR toxicity in this fragment model (*Figure 10C–F*), suggesting that it can play a role in transcription-independent pathological pathways in SBMA, such as protein misfolding and aggregation. Of course, these fragment models show ligand-independent toxicity and therefore cannot recapitulate the specific features of SBMA. Furthermore, the ability of AR to bind DNA is known to be important for toxicity in a full-length mutant AR *Drosophila* model of SBMA, suggesting that SBMA may also arise via a mechanism that involves aberrant gene transcription (*Nedelsky et al., 2010*). Consistent with this idea, changes in gene expression have been detected in SBMA mouse models and this is believed to contribute to pathology (*Sopher et al., 2004*; *Ranganathan et al., 2009*; *Katsuno et al., 2010a*; *Mo et al., 2010*; *Minamiyama et al., 2012*). We therefore wondered whether NLK was also able to affect the function of the full-length AR protein, as this may contribute to the molecular mechanism by which NLK affects SBMA in vivo. We started by testing if NLK could affect the ability of the mutant AR to activate gene transcription by making use of an AR-responsive luciferase reporter. Both wild-type and mutant AR (*Figure 11A* and *Figure 11—figure supplement 1*) are able to activate the expression of this reporter when expressed in DHT-treated NSC-34 cells, although, as expected (*Mhatre et al., 1993*; *Thomas et al., 2006*), AR120Q showed less activity than AR25Q. When NLK was co-expressed with AR, it led to a robust increase in AR-mediated gene transcription in a hormone- and kinase activity-dependent manner (*Figure 11A* and *Figure 11—figure supplement 1*). This effect was also seen with wild-type AR (*Figure 11—figure supplement 1B*), suggesting that NLK may normally act as an AR cofactor or regulator.

We next wondered how exactly NLK was able to influence AR-mediated gene transcription. While most nuclear hormone receptors regulate gene transcription primarily via the interaction of their ligand binding-induced AF-2 domain with cofactors that contain a LxxLL motif, AR is unique in that it contains an LxxLL-like site in its N-terminus ([23]FQNLF[27]) that interacts with its own AF-2 domain with a greater affinity than other motifs (*He et al., 2001*). This intramolecular interaction is known as the N/C interaction and it causes AR to regulate gene transcription primarily through its AF-1 domain in lieu of the AF-2 domain (*He et al., 2001*, *2004*). Loss of this interaction leads to a decrease in AR-mediated gene transcription at some, but not all AR-dependent genes (*He et al., 2002*; *Callewaert et al., 2003*). Interestingly, it has been previously reported that the N/C interaction is upstream of mutant AR aggregation and toxicity, as well as its phosphorylation at both S81 and S308 (*Orr et al., 2010*). Therefore, we predicted that NLK might be acting to promote this intramolecular interaction and thereby increase AR-mediated gene transcription, AR phosphorylation, and SBMA phenotypes. We tested this idea by performing a mammalian two-hybrid assay in which a VP16 activation domain-fused AR N-terminus (VP16-AR120Q-N) is co-transfected with a Gal4 DNA binding domain-fused AR C-terminus (Gal4-AR-C). When these AR N- and C-terminal fragments interact, they bring together the Gal4-DBD and the VP16 activation domain, leading to an increase in the expression of a co-transfected Gal4-dependent luciferase reporter (*Figure 11B*). When we carried out this assay in the presence of NLK, we found that, surprisingly, NLK inhibits the N/C interaction (*Figure 11B* and *Figure 11—figure supplement 2A*). This inhibition was NLK dose dependent (*Figure 11—figure supplement 2B*). The N/C interaction was also inhibited by NLK-KN,

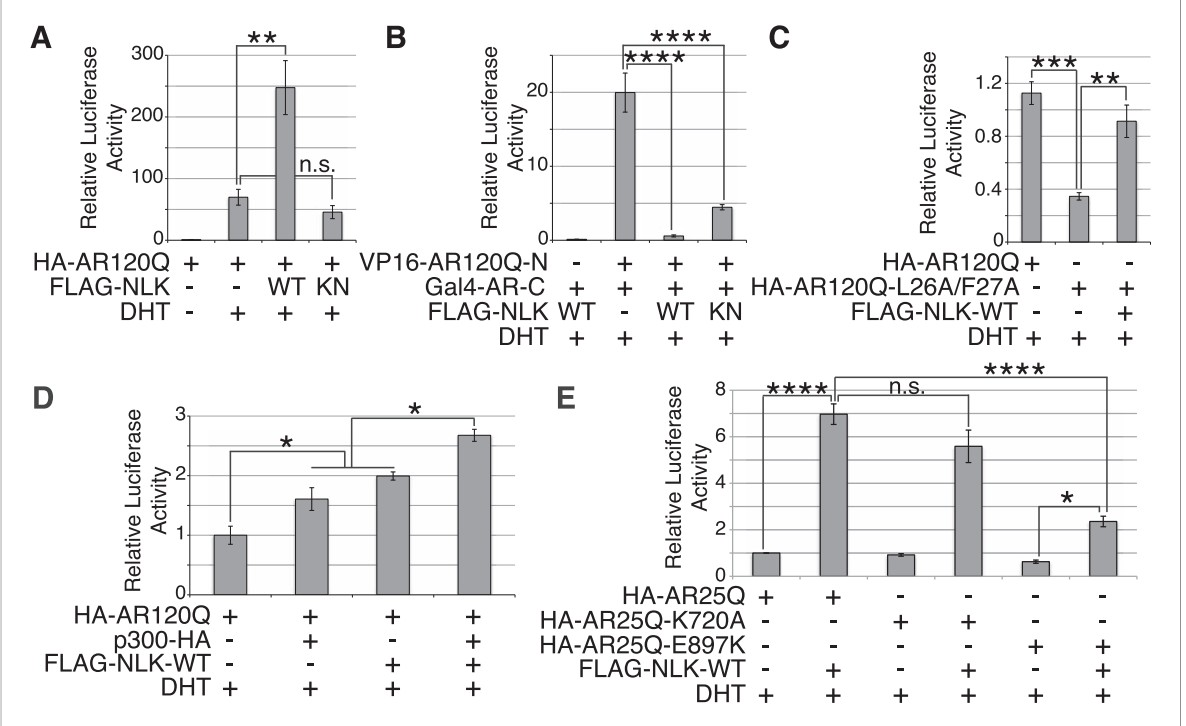

**Figure 11**. NLK promotes AR-mediated gene transcription by inhibiting the N/C interdomain interaction and promoting AF-2 cofactor binding. (**A**) NLK increases AR-dependent gene transcription in a kinase activity-dependent manner in NSC-34 cells. n.s. = not significant, **p < 0.01 (*ANOVA* with Tukey's *post-hoc* analysis). n = 3 trials. (**B**) NLK inhibits the AR N/C interaction as measured by a mammalian two-hybrid assay in NSC-34 cells. ****p < 0.0001 (*ANOVA* with Tukey's *post-hoc* analysis). n ≥ 4 trials. See also *Figure 11—figure supplement 2*. (**C**) NLK can activate AR-dependent gene transcription in the absence of the N/C interaction in NSC-34 cells. **p < 0.01, ***p < 0.001 (*ANOVA* with Tukey's *post-hoc* analysis). n ≥ 3 trials. (**D**) NLK and p300 synergistically increase AR-mediated gene transcription in NSC-34 cells, suggesting NLK may promote AR-cofactor binding and function. *p < 0.05 (*ANOVA* with Tukey's *post-hoc* analysis). n = 5 trials. (**E**) NLK increases AR-mediated gene transcription via the AR AF-2 domain in NSC-34 cells. n.s. = not significant, *p < 0.05, ****p < 0.0001 (*ANOVA* with Tukey's *post-hoc* analysis). n = 4 trials.

The following figure supplements are available for figure 11:

**Figure supplement 1**. NLK does not induce AR transactivation in the absence of hormone.

**Figure supplement 2**. NLK dose-dependently inhibits the AR N/C interaction.

**Figure supplement 3**. NLK can increase mutant AR aggregation and phosphorylation independent of an N/C interaction.

but to a lesser extent (*Figure 11B* and *Figure 11—figure supplement 2A*). Once again, these effects were seen with both wild-type and mutant AR, although NLK inhibited the N/C interaction more robustly in the presence of the polyQ expansion (*Figure 11—figure supplement 2B*). This suggests that NLK is able to prevent the N/C interaction via a mechanism that is only partially dependent on its kinase activity.

In order to confirm that NLK was still able to induce its effects on the mutant AR in the absence of this AR N/C interaction, we introduced a mutation in the N-terminal [23]FQNLF[27] motif of AR that prevents it from binding the AF-2 domain in the C-terminus of the protein (HA-AR120Q-L26A/F27A) (*He et al., 2001*). As previously reported (*He et al., 2002*; *Callewaert et al., 2003*; *Orr et al., 2010*), this construct tended to be compromised in its ability to aggregate (*Figure 11—figure supplement 3A,C*) and was significantly impaired in its ability to induce AR-mediated gene transcription (*Figure 11C*). It also showed a reduction in phosphorylation at AR-S81 (*Figure 11—figure supplement 3D*), as was reported in a separate SBMA cell model (*Orr et al., 2010*). Nonetheless, co-expression of NLK-WT still increased the aggregation rate of this mutant AR (*Figure 11—figure supplement 3A–C*). NLK increased

AR-mediated gene transcription when co-expressed with the AR N/C mutant (*Figure 11C*). NLK was also able to increase the phosphorylation of this construct at S81 (*Figure 11—figure supplement 3D*), as well as at S308, although to a lesser extent (data not shown). Taken together, these data indicate that NLK can influence the activity and toxicity of the mutant AR via a mechanism that is independent, but perhaps parallel to, the AR N/C interaction.

## NLK promotes AR transactivation via the AR AF-2 domain

One important remaining question is how NLK can increase AR-mediated gene transcription while inhibiting its N/C interaction. AR regulates target gene transcription by interacting with several cofactors at both its AF-1 and AF-2 domains (*Bennett et al., 2010*), and these interactions can be altered by polyQ expansion. For example, the coactivator CREB-binding protein (CBP) binds the polyQ-expanded AR more robustly than its wild-type counterpart in a mouse model of SBMA (*Sopher et al., 2004*) and can be sequestered into mutant AR aggregates (*McCampbell et al., 2000*), suggesting that the interaction of this protein with mutant AR may be important for disease pathogenesis in vivo. In addition, AR is acetylated by CBP/p300 (*Fu et al., 2000*), and this acetylation is also important for SBMA pathogenesis (*Montie et al., 2011*). Therefore, we wondered whether NLK regulates AR transcriptional activity by altering cofactor interactions, and chose to look specifically at p300. As expected, we found that co-expression of p300 with NLK showed a higher level of AR-dependent transcriptional activity than with either cofactor alone (*Figure 11D*). This suggests that NLK may enhance coactivator recruitment to the polyQ-expanded AR and thereby increase AR-mediated gene transcription.

Based on our mammalian two-hybrid data (*Figure 11B*), we speculated that the binding of NLK to the N-terminus of AR (*Figure 1C*) sterically blocks the ability of the AR N-terminus to bind the C-terminal AF-2 domain. As the N/C interaction can inhibit cofactor binding at the AR AF-2 domain (*He et al., 2001*), we reasoned that NLK may be acting to relieve this inhibition and thereby promote gene transcription via the AR AF-2 domain. To test this, we introduced two different point mutations into the AR AF-2 domain to differentially inhibit cofactor binding. The AR AF-2 domain is flanked by two charged clamp residues that mediate its interaction with cofactors containing LxxLL or FxxLF motifs. K720A is a partial AF-2 mutation that neutralizes the charge of one of the clamps, preventing LxxLL motif binding and reducing FxxLF motif binding by 50% (*Dubbink et al., 2004*; *Nedelsky et al., 2010*). E897K is a complete AF-2 mutation that reverses the charge at the other clamp, abolishing both LxxLL and FxxLF motif binding (*Dubbink et al., 2004*). We carried out the AR-responsive luciferase assay with both mutants and found that the E897K mutation alone tended to slightly decrease AR-mediated gene transcription compared to that seen with a wild-type AR, while the K720A mutation did not affect AR activity in NSC-34 cells (*Figure 11E*). This is consistent with what was reported in COS-1 cells (*Nedelsky et al., 2010*). When we co-expressed a wild-type NLK with these AR mutants, we were still able to detect an increase in AR-mediated gene transcription with the K720A mutation. In contrast, NLK-mediated enhancement in AR activity was dramatically compromised by the E897K mutation (*Figure 11E*). These data suggest that the NLK-induced increase in AR transcriptional activity is dependent on a functional AR AF-2 domain.

## Discussion

SBMA is a devastating neuromuscular disease without any cure or effective therapy to date. In this study, we explored whether and how NLK could modulate the pathogenesis of SBMA. By utilizing a variety of model systems, we clearly show that NLK is a key regulatory factor capable of modulating AR activity and SBMA pathology. Using a combination of cell culture, *Drosophila*, and mouse models, we show that reduced expression of NLK suppresses, while increased expression exacerbates, mutant AR-associated SBMA pathology, including protein aggregation, cellular toxicity and degeneration, and animal lethality phenotypes. It is particularly intriguing that the effects of NLK on the mutant polyQ-expanded AR and SBMA are consistent across different model systems, as this suggests that the role of NLK in SBMA pathogenesis is fundamental. Furthermore, all of these effects are clearly dependent on the kinase activity of NLK. Our work therefore strongly suggests that a reduction in NLK expression or enzymatic activity could be beneficial for SBMA patients.

Of particular importance is our finding that a 50% reduction in NLK expression partially rescues the phenotypes of an SBMA mouse model (*Figures 6–8*). This improvement in pathology was seen at

20 weeks of age in these mice and was more robust at the later time point of 30 weeks. By very late time points (i.e., 40 weeks), however, a reduction in NLK expression resulted in an improvement of only some of SBMA phenotypes assayed (*Figures 6, 8E*), and did not show a robust effect in other assays (*Figures 7, 8G*). This suggests that a reduction in NLK expression may act to delay disease progression in this model, but is not sufficient to completely prevent the onset of the full SBMA phenotype. It should be noted that the majority of *BAC fxAR121* mice die before reaching this final time point, however, and so we cannot rule out the possibility that the small cohort of mice analyzed at 40 weeks represent an 'escaper' subset of SBMA mice that are slightly healthier than the average *BAC fxAR121* mouse. The reasons for the variation in the SBMA phenotype in these mice are not known, but may be interesting to investigate in the future. It is also worth mentioning that we analyzed the mice on a C57/129 F1 hybrid genetic background. Although they can be considered to be on a pure background for the purpose of this study, we cannot rule out the possibility of mouse background effects. Future studies on a different pure genetic background and/or using NLK inhibitors would be useful in corroborating our findings.

In this study, we uncovered some molecular mechanisms that we predict underlie the role of NLK in SBMA at the cellular level. First, NLK interacts with mutant AR at the N-terminal region of the protein, and, interestingly, polyQ expansion results in a more robust interaction between NLK and mutant AR in comparison to wild-type AR (*Figure 1A–C*). Second, consistent with our data that NLK modulates SBMA features in a kinase activity-dependent manner, NLK promotes the phosphorylation of AR, either directly or indirectly, at multiple sites, including S81 and S308 (*Figure 9*). NLK-induced changes in AR-S81 phosphorylation can be detected in vivo in mice, and AR-S81 phosphorylation likely contributes to the effect of NLK on SBMA pathology in cell culture and *Drosophila* models (*Figure 10*). Interestingly, however, the S81A mutation decreased, but did not completely abolish, the NLK-mediated effects on mutant AR aggregation (*Figure 10A*). This indicates that, while AR-S81 is likely an important NLK phosphorylation site in the N-terminal region of AR, there may be other NLK target sites outside of this region that also contribute to NLK-dependent AR toxicity in SBMA. Finally, NLK can affect the transcriptional activity of the mutant AR protein, and, once again, this is dependent on its kinase activity (*Figure 11*). Unlike aggregation, which is dependent on the presence of a polyQ expansion, this effect is seen with both wild-type and mutant AR. This suggests that NLK normally acts as an AR cofactor or regulator. PolyQ expansion, while resulting in the aggregation of the protein, also affects the activity of the AR monomer, whose altered function in target gene transcription may exert pathology in SBMA (*Nedelsky et al., 2010*). The ability of NLK to promote the activity of the mutant AR could therefore exacerbate this polyQ-induced protein dysfunction. We suspect that both the modulation of mutant AR aggregation and the misregulation of its native functions ultimately contribute to SBMA pathology, and our data suggest that NLK influences both of these pathomechanisms (*Figure 12*).

The binding of NLK to AR, and likely its subsequent phosphorylation, strongly inhibit the AR N/C interaction, and yet paradoxically increase AR-mediated gene transcription (*Figure 11A–C*). This led us to investigate whether NLK could regulate AR activity via the AF-2 domain, and indeed, we found that complete inhibition of cofactor binding at the AR AF-2 domain strongly compromised the ability of NLK to increase AR transcriptional activity (*Figure 11E*). The NLK effect on AR activity was not completely abolished by this mutation, however, suggesting that NLK may promote AF-1-dependent transcription, as well. Furthermore, the E897K mutation compromised the effect of NLK on AR activity more so than the K720A mutation (*Figure 11E*). As these mutations both completely abolish LxxLL motif binding, this suggests that NLK may preferentially allow for FxxLF motif-containing cofactor binding at the AR AF-2 domain, and this possibility warrants further investigation. Once again, as the effect of NLK on AR transactivation and on the AR N/C interaction are seen with both wild-type and mutant AR, this role for NLK in the regulation of AR AF-2 cofactor interactions is likely a normal function of NLK in AR signaling. Given that NLK promotes SBMA pathology, this may suggest that the AR AF-2 domain is important for SBMA pathogenesis. Interestingly, a separate study found that the retinal degeneration phenotypes in a full-length mutant AR *Drosophila* model were also dependent upon the AF-2 domain of AR. This study also found that the E897K mutation at the AF-2 domain led to a more robust rescue of mutant AR phenotypes than K720A, again demonstrating that the ability of the AR AF-2 domain to bind FxxLF-containing cofactors may be important for SBMA pathogenesis (*Nedelsky et al., 2010*).

We also find it intriguing that the effect of NLK on AR molecular functions is very similar to that of the primate-specific Melanoma Antigen Gene Protein 11 (MAGE-11). MAGE-11 has been reported to

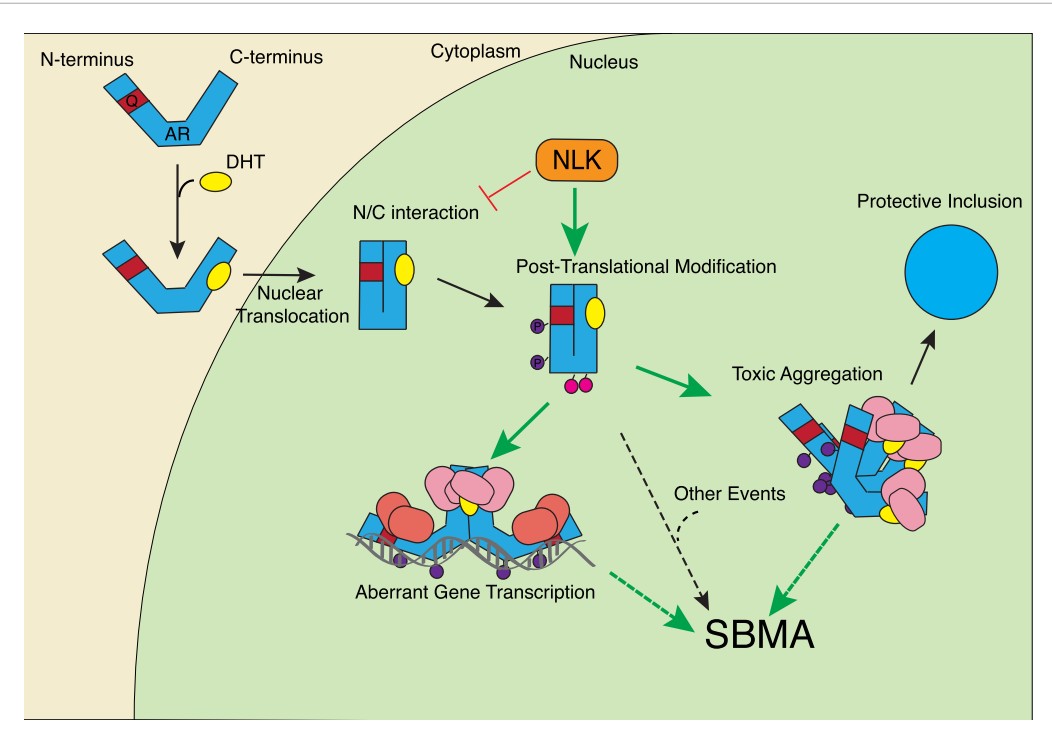

**Figure 12**. A potential model for the role of NLK in SBMA pathogenesis. NLK can induce the phosphorylation of the polyQ-expanded AR, which influences its aggregation and contributes to its toxicity in SBMA models. NLK can also regulate the ability of the mutant AR to act as a transcription factor, which would enhance any aberrant AR-mediated gene transcription that contributes to SBMA pathology. A combination of these toxic mechanisms and others could ultimately result in the degeneration and pathology characteristic of SBMA. These events occur downstream of AR ligand binding and nuclear translocation. In addition, NLK may inhibit the AR N/C interaction to promote AR AF-2 cofactor binding.

bind AR specifically at the [23]FQNLF[27] motif in the N-terminal region of the AR protein to prevent the N/C interaction and allow for cofactor binding at the AF-2 domain (*Bai et al., 2005*). MAGE-11 is also known to directly bridge interactions between AR and various cofactors, including TIF2 and p300, resulting in a synergistic activation of AR-dependent gene transcription (*Askew et al., 2009*, *2010*). Together with our data, this suggests that inhibition of the N/C interaction by specific AR cofactors represents a unique and intriguing approach to regulating AF-domain dominance in AR target gene transcription. It will be interesting to investigate whether there is any cross-talk between NLK and MAGE-11 in AR-mediated gene activation and, perhaps, even in SBMA disease pathogenesis.

We suspect that the NLK-mediated increase in AR transactivation results from an increase in cofactor binding at the AR AF-2 domain, thereby supporting a model in which AF-2-mediated interactions are important for SBMA pathogenesis. And yet, it is also clear that inhibiting the N/C interaction via point mutations in the AR [23]FQNLF[27] motif reduces mutant AR aggregation and toxicity (*Figure 11—figure supplement 3A–C* and *Orr et al., 2010*), features that NLK clearly promotes. One explanation for this seemingly conflicting data is that the binding of NLK to AR and the subsequent phosphorylation of the AR protein, perhaps at S81, elicit an effect on the AR protein that is similar to the effect of the N/C interaction. In this model, NLK binding and the N/C interaction are parallel means of triggering a similar downstream pathogenic response. It should be stressed, however, that binding and phosphorylation by NLK does not preclude the need for AR ligand binding, as aggregation (*Figure 1G* and *Figure 1—figure supplement 1*), gene transcription (*Figure 11—figure supplement 1A*), and the formation of the AR AF-2 domain (*Wärnmark et al., 2003*) all depend upon the presence of androgens, and NLK has no effect on these features without ligand. Furthermore, the exact details of

this downstream pathway are still not completely clear. For instance, toxicity could arise from aberrant AR-mediated gene transcription (via a combination of AF-1 and AF-2 dependent mechanisms), the sequestration of various cofactors into aggregates, the inability of certain cells to handle the accumulation of toxic AR conformers, or via some other as-yet-unknown pathogenic factors. Or, perhaps more likely, SBMA may arise from a combination of the above (*Figure 12*).

Given that NLK interacts with and phosphorylates the mutant AR (*Figures 1, 9*), we suspect that it is acting cell autonomously to regulate AR activity based on the mechanism we propose. As our mouse studies were carried out using a constitutive knockdown of NLK, however, we cannot at this time determine where NLK exerts its effects on SBMA pathology. In other words, it could be regulating mutant AR activity in the spinal motor neurons, the skeletal muscle, or both. Future investigations using targeted NLK inhibitors or tissue-specific knockdown using both this and other SBMA mouse models could address these questions. It is also interesting to note that, although our data show that NLK can influence the aggregation of the mutant AR across multiple models, and that NLK has a robust effect on AR transactivation activity in cells, we saw only a partial rescue of SBMA muscle and motor neuron pathology with a reduction in NLK expression in mice. Why we did not see a more robust improvement of these phenotypes is an intriguing question. A simple explanation may be that the remaining 50% of NLK expression, AR phosphorylation, and mutant protein aggregation is enough to allow for the toxic effects of the mutant AR that directly result in decreased muscle fiber and motor neuron size. A more complete knockout of NLK would thus be needed to prevent degeneration. Mice heterozygous for the *Nlk* gene trap allele are largely normal, suggesting either that this lower expression of NLK is enough to adequately carry out wild-type functions of NLK in the adult mouse, or that some other factor or pathway compensates for the decrease in active NLK. It is possible that such a compensatory factor or pathway may also contribute to mutant AR-induced pathology when NLK expression is reduced.

Data presented here and in a previous publication (*Ju et al., 2013*) suggest that NLK is able to regulate the pathogenesis of two separate polyQ diseases: SBMA and SCA1. In both cases, evidence suggests that NLK binds to and phosphorylates the mutant protein and thereby regulates its aggregation and activity. Yet, why NLK interacts with multiple polyQ proteins is an open question that warrants future investigation. We also noted that co-expression of NLK seemed to increase AR protein levels in NSC-34 cells and in the AR61Q *Drosophila* model, as well as separately influence its propensity to aggregate. This suggests that NLK may play a role in the stabilization of AR, specifically at the protein level, as both of these systems express AR under the control of exogenous promoters. In the *BAC fxAR121* mice, however, the AR121Q protein levels between mice with full NLK expression and those with a 50% reduction in NLK are not significantly different across the population assayed (data not shown). As all the mice assayed did show a rescue in the degenerative phenotype, however, we concluded that another mechanism must be playing a role in these mice and therefore investigated the possibility of a direct interaction between NLK and AR. That direct mechanism is the focus of the current study. Nonetheless, we noted that a subset of about 30–40% of the mice did show a reduction in mutant AR protein levels with a reduction in NLK expression. We therefore speculate that NLK may also play a role in protein clearance pathways, and that this, in turn, may contribute to the ability of NLK to regulate mutant protein aggregation and toxicity in varying disease cases. We ultimately suspect that both direct and indirect regulation of mutant protein expression/aggregation and activity underlies the role of NLK in disease.

Lastly, although there is still much to be understood about the precise molecular mechanisms underlying SBMA and the role of NLK therein, our data clearly show that NLK normally promotes the disease condition and that reduction of NLK expression or activity is sufficient to partially rescue SBMA pathogenicity. We are confident in this conclusion because we utilized a multi-system approach to address the question. NLK is therefore a novel and interesting putative therapeutic target. We should note that complete loss of NLK function may cause severe problems, however, since NLK plays a role in multiple signaling pathways (*Ishitani et al., 1999*; *Ohkawara et al., 2004*; *Ishitani et al., 2010*; *Ishitani and Ishitani, 2013*). Nonetheless, *Nlk^{gt/+}* heterozygous mice are generally healthy and our study provides convincing evidence that a 50% reduction in NLK protects against SBMA pathogenesis in vivo (*Figures 6–8*). Thus, this study suggests that putative treatments that target NLK may not need to completely inactivate the protein to generate a therapeutic effect. It will be very interesting to determine if pharmacologically inhibiting NLK can also rescue SBMA features at the mammalian level.

## Materials and methods

### *Drosophila* genetics

The following mutant and transgenic flies were used in this study: *GMR-Gal4* (Bloomington Stock Center), *UAS-EGFP* (Bloomington Stock Center), *UAS-AR14Q* (current study), *UAS-AR61Q* (current study), *UAS-trAR*[112Q] (*Chan et al., 2002*), *UAS-NLK-WT* (*Ju et al., 2013*), *UAS-NLK-KN* (*Ju et al., 2013*), *nmo*[adk1] (*Verheyen et al., 2001*), *nmo*[adk2] (*Verheyen et al., 2001*). In order to generate *UAS-AR14Q* and *UAS-AR61Q* transgenic fly lines, full-length human *AR* cDNAs with 14Q or 61Q were subcloned into the pUAST vector and then injected into fly embryos (via Best Gene, Inc. Chino Hills, CA). After crossing with the *GMR-Gal4* driver line, two independent *UAS-AR* lines of each Q length that showed roughly equal levels of transgene expression were used for the analysis. For the genetic interaction analyses, appropriate fly lines were intercrossed and their progeny were raised at 22°C, 25°C, or 30°C on fly food containing or lacking 100 nM DHT. All experiments were carried out multiple times.

### Mouse husbandry and genetics

The Yale University Institutional Animal Care and Use Committee approved all research and animal care procedures. Mice were maintained on a 12/12-hr light/dark cycle with standard mouse chow and water ad libitum. Two independent *Nlk* gene trap (*Nlk*[RRJ297/+] or *Nlk*[XN619/+], or simply *Nlk*[gt/+]) mouse lines were maintained on the pure 129S6/SvEv background (*Ju et al., 2013*). *BAC fxAR121* SBMA transgenic mice were maintained on the pure C57BL/6J background (*Cortes et al., 2014*). To perform the genetic interaction study, *BAC fxAR121*[+/−] heterozygote mice were bred to *Nlk*[gt/+] heterozygote mice. The F1 male progeny (C57/129 hybrid background) were used in subsequent analyses.

### Mouse survival analysis

Mice were monitored for their general health and the date of death was recorded. Occasionally, mice were euthanized for humane reasons at the very end stage of disease progression, and the date of euthanasia was used as the death date in the analysis. Survival curves were generated using Kaplan–Meier statistical analysis and the log rank test was used to compare individual curves. The assay was capped at 2 years of age.

### Mouse muscle histology

Mouse quadriceps were harvested and snap frozen in liquid nitrogen-chilled isopentane. Samples were sectioned on a cryostat at 12 μm and collected on superfrost slides. Sections were then either stained with hematoxylin (3 min) and eosin (1 min) or incubated with 0.4 mg/ml NADH (Roche) and 0.8 mg/ml 4-nitro blue tetrazolium chloride (NBT; Roche) for 15 min, 37°C. Sections were then dehydrated with ascending ethanol solutions and incubated in xylenes. Coverslips were mounted with Permount. Slides were imaged on a compound light microscope using an Olympus camera and CellSens software. Fiber area and Feret's diameter of cross-sectional muscle fibers and the mean gray value of NADH transferase activity staining images were analyzed using ImageJ software (National Institutes of Health). The NADH transferase activity images were obtained on the same day using identical camera settings.

### Mouse spinal cord histology

Mouse vertebral columns were dissected whole from freshly sacrificed mice and post-fixed in 4% paraformaldehyde overnight, 4°C. Samples were kept at 4°C through subsequent steps until freezing. After fixing, samples were incubated in 0.5 M EDTA in PBS overnight. The following day, the 0.5 M EDTA was replaced with fresh solution three times, rocking, with the last incubation lasting overnight. The next day, samples were moved to 10% sucrose, then 20% sucrose, and finally left in 30% sucrose overnight. Spinal cord and bone were frozen in Optimal Cutting Temperature (OCT) medium and later sectioned on a cryostat at 18 μm and collected on Superfrost Plus slides. After sectioning, the L4–L5 region was identified based on location and morphology as compared to a mouse spinal cord atlas. Alternating sections were stained with Cresyl violet (4 min) and dehydrated in ascending ethanol solutions. Slides were incubated in xylenes, and coverslips were mounted with Permount. The entire L4–L5 region was imaged on a compound light microscope using an Olympus camera and CellSens

software, and then random images periodically spaced throughout this region were used for the measurement of neuronal soma size using ImageJ. Over 100 neurons were scored per animal.

## Filter trap assay

Quadriceps extracts were generated as for immunoblot and prepared as 400 µl (1 µg/µl) samples. Samples were then divided into 2 equal halves and ran separately though the filter trap assay using a BioRad BioDot SF apparatus according to the manufacturer's instructions, with the exception that a 0.22-µm cellulose acetate (CA) membrane (Whatman) was placed atop the 0.45-µm nitrocellulose (NC) membrane. The CA membrane collects insoluble AR, while the NC membrane detects soluble protein. For one sample half, both the CA and NC membranes were blocked and immunoblotted with anti-AR H280 antibody (1:500, Santa Cruz). For the other sample half, the NC membrane was immunoblotted for the loading control using either rabbit anti-Actin (1:10,000; Sigma) or mouse anti-Tubulin (1:30,000; Developmental Studies Hybridoma Bank). The amount of AR collected by each membrane was quantified using ImageJ.

## Plasmid construction

To generate HA-tagged *AR* constructs, the full-length human *AR* cDNAs were PCR-amplified from *GFP-AR25Q* or *GFP-AR120Q* plasmids and inserted into an HA vector using the XhoI and NotI sites. The 130 amino acid N-terminal fragment was also subcloned into HA and GFP vectors via restriction digest and Gateway cloning, respectively. All *AR* point mutations used in this study were introduced via site-directed mutagenesis using the Stratagene Quikchange Kit. The FLAG-tagged *Nlk* constructs were kindly provided by Dr Kunihiro Matsumoto (Nagoya University, Nagoya, Japan) and Dr Tohru Ishitani (Kyushu University, Fukuoka, Japan). *ARE*-luciferase plasmids were kindly provided by Dr Nancy L Weigel (Baylor College of Medicine, Houston, Texas, USA) and Dr Zafar Nawaz (University of Miami, Miami, Florida, USA). The mammalian two-hybrid constructs were kindly provided by Dr Diane E Merry (Thomas Jefferson University, Philadelphia, Pennsylvania, USA).

## Cell culture experiments

A mammalian cell culture system was used for co-immunoprecipitation and biochemical analyses, immunofluorescence, and luciferase reporter assays. Standard cell culture and plasmid transfection were conducted as described (*Ju et al., 2013*; *Kim et al., 2013*). Briefly, NSC-34 or HeLa cells were maintained in Dulbecco's Modified Eagle Medium (DMEM) supplemented with 10% fetal bovine serum (FBS; Gibco). Cells were plated the day before transfection in 6- or 24-well plates. The following day, cells were transfected with indicated cDNA plasmids using lipofectamine 2000 (Invitrogen) according to the manufacturer's instructions, treated with 10 nM DHT (Wako; dissolved in ethanol) using DMEM supplemented with charcoal:dextran stripped FBS (Gemini Bio-Products), and cultured until analyzed.

## Co-immunoprecipitation (co-IP) assay

To generate cell culture extracts, NSC-34 or HeLa cells were transfected and treated with DHT as described above. 24 hr after DHT treatment, cells were lysed in 300 µl NP40 lysis buffer (0.5% NP40, 20 mM Tris-HCl pH 8.0, 150 mM NaCl, 1 mM EDTA + Protease Inhibitor Cocktail [Roche]). Extracts were cleared by centrifugation for 10 min at 4°C, and soluble extract was either boiled with sample buffer to generate 'input' samples or incubated overnight with either anti-FLAG M2 Affinity Gel (Sigma) or glutathione-sepharose 4B beads (GE Healthcare) as indicated. IP samples were washed three times with lysis buffer and submitted to western blot analysis.

## Western blot analysis

For non-co-IP blots, cells were transfected and treated with DHT and cell extracts were generated as described using triple cell lysis buffer (0.5% NP40, 0.5% Triton X-100, 0.1% SDS, 20 mM Tris-HCl pH 8.0, 180 mM NaCl, 1 mM EDTA + Protease Inhibitor Cocktail [Roche]). Extracts were boiled with sample buffer and ran on 8 or 12% SDS-PAGE gels. Gels were transferred to NC membranes, blocked and incubated with primary antibodies overnight in non-fat milk at 4°C. Membranes were then washed and probed with horseradish peroxidase-conjugated secondary antibodies (GE Healthcare) and exposed to film. Mouse tissue samples were harvested and lysed in 1 ml RIPA buffer (1% NP40, 0.5% sodium

deoxycholate, 0.1% SDS, 50 mM Tris-HCl pH 8.0, 150 mM NaCl, + Protease Inhibitor Cocktail [Roche]) by dounce homogenization and cleared by centrifugation for 10 min at 4˚C. Total protein concentration was measured using a BCA assay and equivalent concentrations of protein were ran on SDS-PAGE gels and blotted. For phosho-AR-S81 blots of mouse tissue, SuperSignal Western Femto (Thermo Scientific) was used in order to detect the signal. Adult *Drosophila* heads were collected and ground in 50 μl RIPA buffer and incubated on ice for 15 min. Samples were then spun for 10 min at 13,000 rpm. Supernatant was boiled with sample buffer for 5 min, ran on 8% SDS-PAGE gels, and blotted. Antibodies used include: mouse anti-HA (1:10,000; Sigma), mouse anti-FLAG (1:10,000; Sigma), mouse anti-GAPDH (1:20,000; Sigma), mouse anti-Tubulin (1:30,000; Developmental Studies Hybridoma Bank), rabbit anti-GST (1:10,000; Sigma), rabbit anti-AR N20 (1:500; Santa Cruz), rabbit anti-AR H280 (1:500; Santa Cruz), rabbit anti-AR phospho-S81 (1:500; Millipore), rabbit anti-AR phospho-S308 (1:500; Santa Cruz), mouse anti-polyQ 1C2 (1:1000; Millipore), and rabbit anti-NLK (1:5000; Abcam). Quantification of immunoblots was carried out using ImageJ using loading controls ran on the same SDS-PAGE gel as the samples. Multiple trials were averaged.

## Immunofluorescence and aggregation analysis

Cells were plated unto coverslips and transfected and DHT-treated. 24 hr after DHT treatment, cells were fixed in 4% paraformaldehyde, permeabilized, blocked, and incubated with primary antibodies (1:1000, mouse anti-FLAG and rabbit anti-AR N20, as indicated). They were then washed, incubated with Alexa Fluor conjugated secondary antibodies (1:500, Life Sciences), and mounted onto slides in Vectashield. Immunofluorescence was imaged using a Zeiss spinning disc confocal microscope using Volocity software. All images are composite z-stacks encompassing the entire cell. Cells were scored as containing aggregates or not based on the presence of punctate vs solely nuclear staining. The ratio of aggregate-containing cells out of total cells was recorded and averaged over at minimum three trials. Mouse quadriceps were harvested and snap frozen in liquid nitrogen-chilled isopentane. Samples were sectioned on a cryostat at 12 μm and collected on superfrost slides. Slides were blocked, incubated with rabbit anti-AR H280 (1:200; Santa Cruz), washed and incubated with Alexa Fluor 488 secondary antibody (1:500, Invitrogen) and TOTO-3 (1:1,000, Invitrogen). Coverslips were mounted with Vectashield. Slides were imaged on a Zeiss LSM710 confocal microscope and images are z-stack composites encompassing the entire section. Aggregation rate was determined using ImageJ software: 'Particle Analysis' was used to determine the number of TOTO-3-stained nuclei and the 'Find Maxima' tool was used to locate all AR aggregates. The ratio of nuclei containing aggregates out of total counted was recorded for several sections and averaged over individual mice by genotype.

## Primary motor neuron culture and analysis

Primary motor neurons were prepared from embryonic day 13 (E13) mouse embryos as described previously with slight modification (*Gingras et al., 2007*; *Montie et al., 2009*). Briefly, spinal cords were dissected in ice-cold L15 medium (Gibco), dissociated in 0.05% trypsin, and plated on poly-D-lysine- and laminin-coated plates. After 7 days, cells were transfected with *GFP-AR120Q* and/or *FLAG-Nlk-WT* plasmids by the Calcium-phosphate method. On the next day, 10 μM DHT was added to the medium. At DIV9, cells were fixed and subjected to immunofluorescence. Primary antibodies used were mouse anti-FLAG antibody (1:1000, Sigma), rabbit anti-GFP antibody (1:1000, Abcam), and goat anti-ChAT antibody (1:100, Calbiochem). Appropriate Alexa secondary antibodies (Invitrogen) were used to visualize the proteins. The number of aggregate-containing cells per total GFP-positive cells was counted manually.

## Luciferase assay

NSC-34 cells were transfected with an *ARE-luciferase* reporter, a pRL-TK *renilla luciferase* reporter and any other indicated constructs using lipofectamine 2000 and treated with DHT. 24 hr after DHT treatment, cells were lysed and subjected to a dual-luciferase assay using a Promega kit according to the manufacturer's instructions. Luciferase activity was measured using a Promega GloMax 20/20 luminometer and associated software. The ratio of the luciferase activity values was recorded for each sample and normalized to control samples in each case. Each experimental trial was performed in triplicate, and ratios were averaged over multiple trials. Protein expression was confirmed by immunoblot.

## Mammalian two-hybrid assay

Mammalian two-hybrid assays were carried out as described for the other dual-luciferase assays, except a *Gal4-luciferase* reporter was used in place of the *ARE-luciferase* construct and cells were transfected with the VP16- and Gal4-DBD-fused protein constructs as indicated.

## Statistics

Unless otherwise noted, statistical significance between two sample sets was determined by the Student's *t-test* using a two-tailed distribution and assuming unequal variance. Statistical significance between multiple sample sets was determined by one-way *ANOVA* using Tukey's *post-hoc* HSD test to compare individual group differences. Statistics were calculated using Microsoft Excel and GraphPad Prism software.

## Acknowledgements

The authors would like to thank the members of the Lim Lab for help and feedback on this study. We would also like to thank Dr Nancy M Bonini for supplying the trAR$^{112Q}$ *Drosophila* model, Dr Nancy L Weigel and Dr Zafar Nawaz for the *AR* and *ARE-luciferase* reporter constructs, Dr Diane E Merry for the mammalian two-hybrid constructs and her critical reading of and feedback on this study, and Dr Kunihiro Matsumoto and Dr Tohru Ishitani for the *Nlk* constructs. This work was supported by the National Institute of Neurological Disorders and Stroke grants F31 NS081811 (to TWT), R00 NS064146 and R01 NS083706 (to JL), and R01 NS041648 (to ARL), the Muscular Dystrophy Association (Basic Research Grant to ARL), the Brain & Behavior Research Foundation (Formerly NARSAD), the Alfred P Sloan Foundation, the National Multiple Sclerosis Society, the Charles H Hood Foundation, the National Ataxia Foundation, and the Yale Scholar Award Program (to JL).

## Additional information

### Funding

| Funder | Grant reference | Author |
| --- | --- | --- |
| National Institute of Neurological Disorders and Stroke | F31 NS081811 | Tiffany W Todd |
| Brain and Behavior Research Foundation | Young Investigator Award | Janghoo Lim |
| Alfred P. Sloan Foundation | Sloan Research Fellowship | Janghoo Lim |
| National Multiple Sclerosis Society | Pilot Grant | Janghoo Lim |
| Muscular Dystrophy Association | Basic Research Grant | Albert R La Spada |
| Charles H. Hood Foundation | Child Health Research Award | Janghoo Lim |
| National Ataxia Foundation | Research Grant | Janghoo Lim |
| Yale University | Yale Scholar Award | Janghoo Lim |
| National Institute of Neurological Disorders and Stroke | R00 NS064146 | Janghoo Lim |
| National Institute of Neurological Disorders and Stroke | R01 NS083706 | Janghoo Lim |
| National Institute of Neurological Disorders and Stroke | R01 NS041648 | Albert R La Spada |

The funders had no role in study design, data collection and interpretation, or the decision to submit the work for publication.

### Author contributions

TWT, JL, Conception and design, Acquisition of data, Analysis and interpretation of data, Drafting or revising the article; HK, HCM, CJC, ARLS, Conception and design, Acquisition of data, Analysis and interpretation of data

## Ethics

Animal experimentation: This study was performed in strict accordance with the recommendations in the Guide for the Care and Use of Laboratory Animals of the National Institutes of Health. All of the animals were handled according to approved institutional animal care and use committee (IACUC) protocols (#2013-11342) of the Yale University. The Yale University Institutional Animal Care and Use Committee approved all research and animal care procedures. We made every effort to minimize animal suffering.

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
