## [Decision Letter]

Thank you for submitting your work entitled “Nemo-like kinase is a novel regulator of spinal and bulbar muscular atrophy” for peer review at *eLife*. Your submission has been favorably evaluated by a Senior Editor and two reviewers, one of whom is a member of our Board of Reviewing Editors.

The reviewers have discussed the reviews with one another and the Reviewing editor has drafted this decision to help you prepare a revised submission.

The following individuals responsible for the peer review of your submission have agreed to reveal their identity: Louis Ptáček (Reviewing editor and peer reviewer) and Hank Paulson (peer reviewer).

Todd and colleagues explore the role of Nemo like kinase (NLK) as a regulator of disease processes in the neurodegenerative disorder, spinal and bulbar muscular atrophy (SBMA). SBMA is a polyglutamine expansion disease that specifically affects motor neurons. Todd and colleagues use a wide range of models and assays to address the potential role of NLK in disease pathogenesis. Particular strengths of the study include genetic crosses between SBMA mice and mice haploinsufficient for NLK that strongly suggest it is indeed a modulator of disease processes and studies establishing that NLK can phosphorylate the disease protein in SBMA, the androgen receptor (AR), at specific sites. Taken together, results in the mouse crosses, fly models of disease and in cell-based assays of transcription, phosphorylation and protein-protein interaction, support the view that NLK directly phosphorylates the androgen receptor and directly binds the protein. While much of the data support the view that AR phosphorylation mediated by NLK is a principal driver of AR protein accumulation, aggregation and toxicity, additional evidence suggests that a direct interaction between AR and NLK independent of kinase activity inhibits AR intramolecular (N/C) interactions and alters AR association with transcription factors, separately adding to the toxicity of expanded AR. Overall the results are thorough and build a solid case for the role of NLK in this polyglutamine disease. While there are some inconsistencies in the data – for example, the variable extent of modulation in two separate fly models deficient in the NLK ortholog and an intriguing discrepancy between established factors contributing to toxicity and the effect of NLK on intramolecular interactions of mutant AR – these are not a significant issue and do not lessen the import of the study. Indeed, the authors do a good job of explaining their, sometimes conflicting, data and avoiding over-interpretation of key findings.

Studies by the same group previously showed that NLK could modulate SCA1 disease pathogenesis, SCA1 being another polyglutamine disorder. While this convergence on NLK in two related diseases arguably increases the impact of the current findings, it is striking that two separate disease proteins with little functional connection show similar relationships to NLK. One wonders whether there are indirect neuroprotective effects of this kinase on cellular pathways, not yet been explored, that may explain aspects of disease regulation by this signaling protein. The studies are a significant advance in the field and provide insight into the broader area of pathogenic factors underlying polyglutamine disorders.

Essential revisions:

1) Figures 1, 2, 3, 4, 5 and 10 ANOVA should be used instead of *t*-test in these multivariable instances.

2) A larger question that needs to be addressed is why a major change seen in the molecular data regarding aggregation and phosphorylation and a relatively minor change is seen phenotypically in the mouse (lifespan, muscle fiber, etc.).

3) The first paragraph of the Discussion belongs in the Introduction.

4) Third paragraph of Discussion: why is the 40-week old data not shown but discussed in the manuscript?

5) In the Discussion section, a hypothesis is floated out to the reader, conformational change, multiple times. This needs to be simplified as no data are presented.

6) What is NLK's role in normal AR function and what about the polyQ causes the degenerative phenotype?

---

## [Author Response]

*1)*
Figures 1, 2, 3, 4, 5 and 10
*ANOVA should be used instead of* t*-test in these multivariable instances*.

We appreciate the reviewers’ drawing our attention to this oversight. The statistics for the figures listed have been updated to *ANOVA* using Tukey’s post-hoc analysis to compare samples. We also updated the figure legends and methods section to reflect these changes. Some of the more subtle effects we noted in our original manuscript are not significant by *ANOVA*, and we have altered the text to reflect this. All major conclusions concerning the role of NLK in SBMA remain the same and in some cases are of increased significance using this method.

*2) A larger question that needs to be addressed is why a major change seen in the molecular data regarding aggregation and phosphorylation and a relatively minor change is seen phenotypically in the mouse (lifespan, muscle fiber, etc.)*.

We apologize that we did not directly address this question in our original manuscript. We agree that it is an important question and have included a discussion of our thoughts on this topic in the Discussion section of the revised manuscript.

*3) The first paragraph of the Discussion belongs in the Introduction*.

We agree with the reviewers. We have moved this paragraph from the Discussion to the Introduction and merged it into the existing text.

*4) Third paragraph of Discussion: why is the 40-week old data not shown but discussed in the manuscript*?

The 40-week data we referred to in the Discussion are included in the graphs in Figures 6, 7 and 8, alongside data from earlier time points. We have now noted this in the text.

*5) In the Discussion section, a hypothesis is floated out to the reader, conformational change, multiple times. This needs to be simplified as no data are presented*.

We agree that our original manuscript included a more lengthy description of our hypotheses than was warranted given our current data. We have edited the Discussion section to focus less on this topic and more on the questions in Essential Revisions #2 and 6 listed in this letter.

*6) What is NLK's role in normal AR function and what about the polyQ causes the degenerative phenotype*?

We agree with the reviewers that this is an important question, and in our revised manuscript we have tried to better explain our thoughts on this topic in our Discussion. To summarize, we currently suspect, based on the data presented, that NLK normally interacts with and phosphorylates AR and regulates its interactions with its cofactors. This in turn influences the transactivation activity of AR. However, others and we have postulated that its activity and its interactions with its cofactors are likely altered, resulting in protein dysfunction and pathology. As NLK is still able to bind this mutant AR, we predict that its normal role in regulating AR activity now acts to enhance the dysfunction and toxicity of the receptor. At the same time, our data suggest that phosphorylation of the mutant (not wild-type) AR by NLK promotes its aggregation, enhancing the toxic effects of this aggregation process, as well.